



# Predicting and reducing wind energy field experiment uncertainties with low-fidelity simulations

Daniel R. Houck[1], Nathaniel B. de Velder[1], David C. Maniaci[1], and Brent C. Houchens[1]

[1]1515 Eubank Blvd SE, Albuquerque, NM 87123

**Correspondence:** Daniel Houck (drhouck@sandia.gov)

**Abstract.** Experiments offer incredible value to science but results must always come with an uncertainty quantification to be meaningful. This requires grappling with sources of uncertainty and how to reduce them. In wind energy, field experiments are commonly conducted with a control and treatment. In this scenario bias errors can usually be neglected as they impact both control and treatment approximately equally. However, random errors propagate such that the error in the difference between the control and treatment is always larger than the random errors in the individual measurements. As random errors are usually reduced with additional measurements, there is a need to know the minimum duration of an experiment required to reach acceptable levels of uncertainty. We present a general method to simulate a proposed experiment, calculate uncertainties, and determine both the measurement duration (total time of measurements) and the experiment duration (total time to collect the required measurement data when including condition variability and time when measurement is not occurring) required to produce statistically significant and converged results. The method is then demonstrated as a case study with a virtual experiment that uses real-world wind resource data and several simulated tip extensions to parameterize results by the expected difference in power. With the method demonstrated herein, experiments can be better planned by accounting for specific details such as controller switching schedules, wind statistics, and post-process binning procedures such that their impacts on uncertainty can be predicted and the measurement duration needed to achieve statistically significant and converged results can be determined before the experiment.

## 1 Introduction

There is a long history of experiments in wind energy and their necessity is still evident today. There have been several recent experiments to test wake steering for example (Fleming et al., 2019), (Simley et al., 2022), (Howland et al., 2022). The ongoing Rotor, Aerodynamics, Aeroelastics, and Wake (RAAW) campaign is exemplary of the need for experiments and field measurements in wind energy as it seeks to provide a new and better validation data set (Kelley et al., 2023), (Letizia et al., 2023), (Rybchuk et al., 2023). The data produced by any experiment is most valuable when accompanied by uncertainty quantification that allows interpretation of accuracy. Oftentimes, our best attempts at an experiment produce results that, with properly calculated errorbars, are, at least in part, not statistically significant (Scholbrock et al., 2015), (Doekemeijer et al., 2020). Results like these are still of great value as they indicate the need to change measurement procedures, increase instrumentation accuracy, or record data for a longer duration.



All experiments may suffer both bias (epistemic) and random (aleatoric) errors. The former is characterized by a non-zero mean and zero variance while the latter has a zero mean and a non-zero variance. Bias errors frequently originate in instrumentation that drifts out of calibration or from the turbine itself in the case of a wind energy field experiment (e.g., a yaw error). Reducing bias errors can be a tedious process to understand their precise sources and address the underlying causes. In wind energy field experiments, as in many disciplines, the interest is often the difference between two scenarios, for example, a controller design for wake or load mitigation (Fleming et al., 2019); a blade design (Bak et al., 2016), (Castaignet et al., 2010), (Couchman et al., 2014), (Gomez Gonzalez et al., 2021); or the effects of different atmospheric conditions (Lange et al., 2001), (Belu, 2012), (Simley et al., 2022). In these control and treatment experiments, it is often safe to assume that the bias errors are negligible to the difference because the non-zero mean of the total bias error is often equal in control and treatment and would be subtracted when considering the difference. While this reduces one source of uncertainty, it also introduces a new complication in that the random error from two sources (the control and the treatment) must now be propagated into the difference. For example, if

$$\Delta P = P_1 - P_2, \tag{1}$$

where $\Delta P$ is the difference between, say, power measurements from the control, $P_1$, and treatment, $P_2$, each of those measurements has some uncertainty from random errors, $\delta P_1$ and $\delta P_2$. If the experimental setups from which the two measurements are taken are the same in every other way, say control and treatment blades mounted on the same turbine, or two of the same turbines in very similar conditions (statistically the same), it may be possible to assume that $\delta P_1 \approx \delta P_2$. In that scenario, when these errors are combined in quadrature as

$$\delta(\Delta P) = \sqrt{(\delta P_1)^2 + (\delta P_2)^2} = \sqrt{2(\delta P_1)^2} \approx 1.4\delta P_1, \tag{2}$$

the error or uncertainty in the difference, $\delta(\Delta P)$, is approximately 40% larger than the individual errors. Represented as errorbars, a significant difference between control and treatment does not have zero within the uncertainty interval.

The above equation can also be solved for a maximum allowable random error to achieve a predicted difference within uncertainty. For example, if a difference of 2% in a quantity of interest (QoI) was expected between the control and treatment, it can be shown that this requires that the random error of the individual measurements be only about 1.4% of the QoI. Wind energy experiments are frequently hoping to measure differences as small as 1-2% (Maniaci et al., 2020), further emphasizing the challenges to reduce uncertainties to sufficiently low levels to produce statistically significant results. Unlike bias errors, random errors can usually be reduced simply by measuring over a longer duration. By measuring over a longer duration, the distribution of random contributions to error is more completely measured and the mean random error driven toward its theoretical value of zero.

Besides ensuring that results are significant, it is also important when considering ensemble statistics to ensure that data have converged to a given standard. When possible, for example in a controlled lab setting, long records can be recorded during stationary inflow conditions and a suitable convergence standard determined from this measurement. In the field, however, stationarity is not guaranteed and there are usually too many combinations of possible inflow conditions to consider. Nevertheless, it is critical to provide some measure of the convergence of each data set after binning and this too can be converted into a





required measurement duration as it again amounts to knowing how many samples are needed in a given bin. Like significance, convergence is also ensured by increasing the number of samples, but the rates at which convergence and significance are achieved may be different.

A key distinction we intend to make is the difference between the measurement duration required to reach significance and convergence and the experiment duration required. If measurements are uninterrupted, then these are equal. Occasionally,

however, turbine operation must be attended, which leaves large portions of time at which there are no measurements, or instrumentation may have restrictions that limit continuous measurements. These situations may require longer experiment durations to capture measurements across the full range of required conditions. The key questions this paper aims to answer are: *What minimum measurement duration is required to achieve a sufficiently small uncertainty in the difference between control and treatment to yield a statistically significant and converged result? And furthermore, what experiment duration is required to*

*achieve the minimum measurement duration?*

Using simulations to prepare for and predict the results of experiments is regular practice. Doekemeijer et al. (2020), Fleming et al. (2019), and Simley et al. (2022) used the FLOw Redirection and Induction in Steady state (FLORIS) model (NREL, 2020) to make predictions about their field experiments. It is rarer to use simulations to determine how much measurement time will be required to produce statistically significant results from a wind energy field study. Petrone et al. (2011) considered

wind turbine performance under the uncertainty of several parameters, but not to aid in a field experiment, and Cassamo (2022) demonstrated algorithms for processing field data with constraints to produce desired uncertainty levels. Toft et al. (2016) came closest to the method proposed herein by evaluating the contributions of different wind parameters to the probability of turbine failure through a suite of OpenFAST simulations with TurbSim inflows.

Herein, we outline a method that can aid in the prediction of minimum measurement durations necessary to produce sta-

tistically significant and converged results in wind energy field experiments specifically with intent to reduce random errors. The method is first outlined very generally to emphasize that it is highly adaptable to many types of experiments and that it is software agnostic within the guidelines provided. Then, the method is demonstrated for an imagined field experiment, informed by real wind-resource data such that several nuances can be better illustrated and explained.

## 2  General Methodology

The method described and demonstrated herein is highly flexible and adaptable to the particular needs of the experiment. At a very high level, it consists of performing a suite of simulations to represent a proposed experiment with a balance between computational time and fidelity. The outputs of the simulations are then used to perform a statistical analysis to quantify uncertainty and convergence to standards determined by the user and this data is finally converted into a prediction of the minimum measurement and experiment durations required to produce significant and converged results. At this level, the

proposed method could be used for a variety of experiments in many fields, though the focus here is on wind energy and, in particular, field experiments as these present a particular challenge with long measurement durations required to reduce random errors.





The simulation method, inflow representation, and uncertainty analyses will be discussed next in general terms and again, with reference to a case study, after.

## 2.1 Simulation Method

First, an appropriate simulation code is needed. Here, "appropriate" has several requirements. First and foremost, it must simulate the quantities of interest (QoI) to be measured in the experiment with acceptable accuracy. Second, it must be fast enough with available resources to run potentially thousands of simulations that cover the wide range of operating conditions possible. This also assumes that validated models of any turbines in the experiment are also available for use in the chosen code. Finally, it requires that the inflow be represented with enough fidelity to simulate the experiment and capture effects of any specific conditions that are expected to be important to the QoIs. High fidelity may not be needed as long as the expected variance is statistically represented.

## 2.2 Inflow Representation

As any wind energy experiment is essentially a response to the inflow, the inflow conditions are the first required input. For a field experiment, this requires knowledge of the wind resource at the site and time of year when the experiment will take place. In contrast, in a wind tunnel experiment or simulation the inflow is typically prescribed or controlled. When simulating a representative inflow for a field experiment, ideally historical data from a meteorological (met) tower at the site can be used to reduce uncertainties and required assumptions about the inflow conditions. If there is not met data, probabilistic distributions of inflow parameters such as hub height wind speed, turbulence intensity, and shear exponent (the specific parameters will depend on the simulation code being used) could be used to construct representative inflows. One difficulty with the latter approach is determining the potential for correlation among parameters such that the joint probabilities are accurately constructed to represent conditions at the site. Temporal (i.e., time of year and day) distributions, as opposed to probabilistic, help with this construction. When using historical data, it is best to use data from the time period of interest (e.g., certain months and/or hours) over multiple years to have a more robust representation of "typical" conditions as individual years may differ.

After selecting the simulation method and having acquired representative inflow data, the inflow data are now processed into the format required by the simulation code. Here, the method uses 10-minute bin intervals, which is standard for wind energy field experiments, though it could be easily adapted for other needs. This accepts that the effects of phenomena happening on shorter time scales could be reduced due to long averages and phenomena happening on longer time scales may not be adequately captured, so this averaging time is an important consideration depending on the goals of the experiment. The simplest approach when using historical data is to create 10-minute bins, calculate the necessary statistics for each bin (e.g., hub height wind speed, turbulence intensity, and shear exponent), and then use those as inputs to create inflows for the simulations. It is likely necessary to apply some level of quality control to the historical data before doing this. Depending on the robustness of the historical data set, it may be necessary to use statistics on bins shorter than 10 minutes to ensure that enough inputs can be created to represent the time period of the experiment. If so, and especially if the bin length is short, it is advisable to check the correlation time of the historical data (assuming time series are available) to ensure that the length of each bin is





longer than the decorrelation time. This ensures that each input for the creation of simulated inflows is unique. Precisely how many unique inflow realizations are needed to accurately represent a given time period is a matter of judgment. If, for example, extreme loads are of interest, then it will be more important to capture extreme inflow conditions and a larger data set may be necessary. For more typical operating conditions, missing the tails of the distributions of conditions may be acceptable.

Once the set of simulated inflows is complete, the simulations are run with outputs for the QoIs. Again, assuming the field experiment standard of 10-minute statistics, each simulation is run to acquire 10 minutes of usable data (i.e., after any start up time) such that each simulation represents one 10-minute bin of field data and statistics from each simulation are calculated for further analysis.

## 2.3 Analysis and Uncertainty Quantification

The analysis stage may vary depending on the experiment and QoI, but the goal of this method is to quantify the uncertainty. Using the mean statistics of each simulation, the data are binned on inflow statistics, most likely by wind speed, though they could be binned on other parameters or even on multiple parameters. In each resulting bin, a running bootstrap analysis is performed (Efron and Tibshirani, 1986). Often, two standard deviations are reported as the uncertainty interval, however this assumes that the underlying distribution is Gaussian. The bootstrap analysis, on the other hand, makes no assumption regarding

the underlying distribution and so offers a more accurate prediction. Specifically, the bootstrap analysis is used to calculate a confidence interval on the running mean of each QoI such that it is updated for each sequential sample that is added to the bin. It is for the user to decide what the appropriate confidence interval is, though we will offer a few words of caution. The P-value, or $\alpha$, which is one minus the confidence interval, that is chosen for an experiment is in most ways arbitrary. In fact, the originator of the idea of a P-value, Ronald Fisher, chose 0.05 as only an example and never intended it to be a definitive test

(Nuzzo, 2014). The calculation of a confidence interval and whether or not the QoI is significant is not sufficient on its own to draw any conclusions (Wasserstein and Lazar, 2016). It merely suggests whether or not the data are more or less compatible with the hypothesis, and further support for a hypothesis is then needed in the form of other statistical evidence. In practice, whether or not a QoI is significant as an outcome of this method could be used to indicate if the actual experiment is even warranted.

A word of caution is also needed because bootstrap analyses are not robust with small sample sizes (Jenkins and Quintana-Ascencio, 2020). While there is no firm agreement in the literature, a minimum number of samples in the range of 8-25 is probably necessary for a meaningful bootstrap analysis, with a higher minimum needed when the data set is known to have higher variance (Jenkins and Quintana-Ascencio, 2020). This helps prevent narrowness bias that the bootstrap method can cause (Hesterberg, 2014). The running bootstrap analysis can start at the selected minimum, or bins with fewer samples than

the minimum can simply be discarded at any stage. There are several recommendations in the literature on the minimum number of resamples necessary, i.e., the number of replicates created by sampling the original data set with replacement. Hesterberg (2014) makes a compelling argument to use a minimum of 15,000 and we follow this. Note also that, with as few as six samples, there are over 46,000 unique permutations when sampling with replacement. The bootstrap calculated confidence interval now quantifies the random error, primarily associated with inflow conditions, of each QoI for each bin for the control





and treatment.

If the experiment is a control and treatment, then, for each QoI and bin, the difference between the control and treatment is found and the errors combined with the root-sum-square, both on a running basis. From this, the significance and convergence criteria can be selected and applied, and the sample number at which these are both achieved in each bin for each QoI can be determined. Finally, the sample number is converted into a record time using either timestamps of the original inflow data or

the probabilistic distribution. If the experiment is a control and treatment, this is all that is required to quantify uncertainty as previously discussed. If it is not, any bias errors should be calculated for each QoI in each bin as needed and then combined with the random error using the root-sum-square before applying significance and convergence criteria (JCGM100:2008, 2008).

### 2.4 General Discussion

As the goal of this method is to determine how long data must be recorded to ensure statistically significant and converged

results, it is critical that the inflow conditions be represented as accurately as possible and that the QoIs be simulated as accurately as possible, though perhaps allowing for some trade-offs in computation time. The results of this procedure really determine a minimum amount of time required as it assumes no additional quality control or filtering are required, i.e., every simulation is assumed valid. Any real experiment will of course have issues with sensors, unexpected delays, etc. that are not accounted for in this procedure, which will increase the required duration of the experiment.

The uncertainty can also be considerably affected by the analysis and in particular the binning process. While more iterative methods of binning can be used after data collection to ensure certain levels of uncertainty are achieved (Cassamo, 2022), this method allows one to test various methods of binning and analysis beforehand and weigh their advantages against the potential requirement for increased data collection time. For example, different bin widths can be tested to determine the effects on number of samples required for convergence and certainty and those results can be converted into a duration of data collection.

Similarly, binning on multiple parameters can be tested. If one turbine is being used as the control and the treatment by, for example, switching between two control methods, the data produced with this method can be analyzed to determine an optimal switching schedule to achieve the desired results.

### 3 Case Study Example of a Blade Tip Extensions

In this example of the method, we imagine an experiment at the Scaled Wind Farm Test (SWiFT) (see Fig. 1) facility operated

by Sandia National Labs in Lubbock, Texas (Berg et al., 2013). At the SWiFT site, there are three modified V27 turbines with 27 m rotor diameters (D) sited such that wind turbine generator a1 (WTGa1) and WTGb1 are ideally situated for control and treatment experiments relative to the dominant wind direction from the south. Each also has a dedicated met tower (METa1 and METb1) 2.5D upstream in the dominant wind direction with sonic anemometers at 18, 31 (hub height), and 45 m.

For the experiment, we imagine operating WTGa1 as the baseline, or control, in a control and treatment experiment. For

WTGb1, we will test five different tip extensions designed only to produce a difference in power over the control. Using



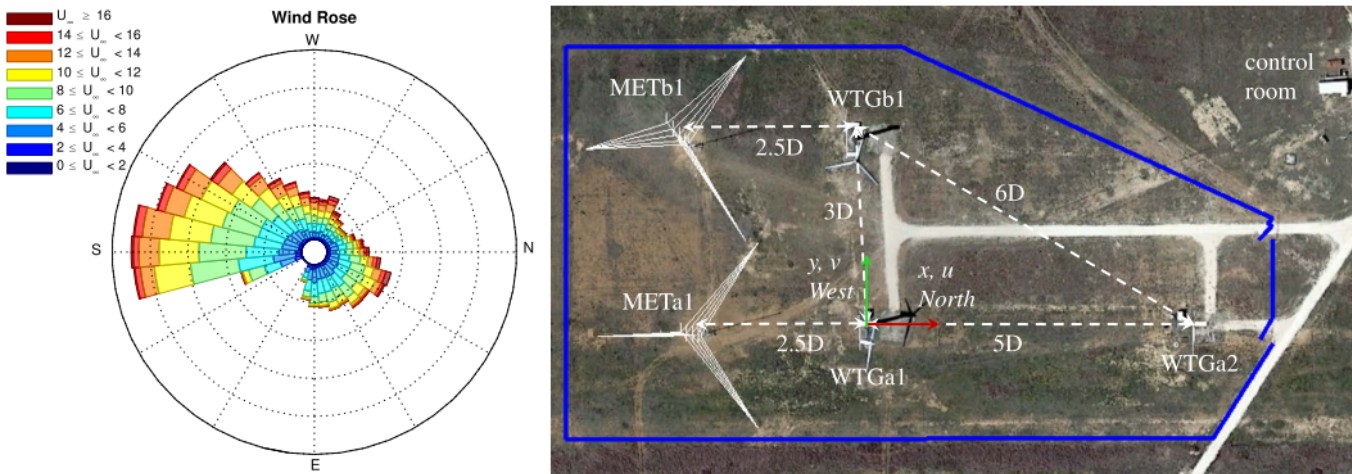

**Figure 1.** The Scaled Wind Farm Test (SWiFT) facility in Lubbock, TX and a representative annual wind rose for the site. Image taken from (Herges et al., 2017). Note both images are oriented with North to the right.

historic data from METa1 and METb1, we can calculate the necessary statistics to represent testing over three months in a suite of OpenFAST simulations using TurbSim inflows.

### 3.1 Tip Extensions

In this virtual experiment, five tip extensions are created to be the treatment rotor and to represent different levels of expected
change between the control and treatment such that the results can be parameterized by the expected change. The design of the tip extensions is based purely on the expected proportion between power and rotor-swept area:

$$D_{treat} = \sqrt{(P_{treat}/P_{ctrl})D_{ctrl}^2},$$ (3)

where $D_{treat}$ is the diameter of a treatment rotor with tip extension, $P_{treat}$ and $P_{ctrl}$ are the desired power of the treatment and control rotors, respectively, and $D_{ctrl} = 27$ m is the diameter of the control rotor. This assumes that all rotors have the same
coefficient of power, which is

$$C_P = \frac{P}{^1/_2\rho A U_\infty^3},$$ (4)

where $P$ is mechanical power, $\rho$ is the air density, $A$ is the rotor-swept area, and $U_\infty$ is the freestream wind speed. Each tip extension is created by linearly extrapolating the chord and twist (the V27 blade has no curve or sweep) of the control rotor and using the same airfoil as the original tip in any new blade stations. These changes are made in the blade definition of
OpenFAST's AeroDyn module. Table 3.1 shows the diameters of all rotors and the expected and actual power changes.



**Table 1.** Diameters, expected and actual increases in power, and $C_P$ for each rotor.

| D [m] | 27 | 27.1 | 27.2 | 27.4 | 27.7 | 28.3 |
|---|---|---|---|---|---|---|
| Expected region 2 power gain [%] | - | 0.75 | 1.5 | 3 | 5 | 10 |
| Actual region 2 power gain [%] | - | 1.98 | 2.67 | 3.95 | 5.63 | 11.12 |
| Actual average region 2 $C_P$ [a] [-] | 0.3966 | 0.4003 | 0.4000 | 0.3991 | 0.3968 | 0.3999 |

[a] Average $C_P$ is calculated using average $P$ and average $U_\infty$ in each wind speed bin as opposed the average of all $C_P$ in a bin.

In addition to modifying the blade properties, each rotor uses the Rotor Open Source Controller (ROSCO) (NREL, 2021)
tuning procedure to ensure that it is operated optimally by finding the region 2 combination of blade pitch and tip speed ratio
that maximizes $C_P$. The rated power is also fixed for all rotors to represent installing these rotors on the same generator. The
goal of these five tip extensions is exclusively to produce a parameterizable difference in operation from the control rotor and
no additional design work was performed. It is sufficient for this demonstration that the rotors create a difference to measure.

It is notable that every tip extension exceeds the estimated difference in power as shown in Table 3.1. This is because the
controller optimization leads to very small differences in $C_P$ among the rotors, which can lead to the relatively larger differ-
ences between the expected and actual power gains. Note that the rotors with a $C_P$ that is closer to the baseline better match the
expected power gain. In a real experiment, we would also expect to modify controller parameters to optimize rotor performance
within limitations such as blade and tower loads. For the purposes of this demonstration, we accept these changes in $C_P$ and
report all findings assuming an optimized controller for each rotor. This demonstrates the importance of measuring differences
due to all modifications of a rotor including physical and operational.

### 3.2  Inflow Creation for the Case Study

As mentioned, the SWiFT site has two met towers, each upstream of the two turbines to be simulated, which allows us to use
historical data to accurately represent inflow conditions at the test site. Additionally, a 200 m met tower operated by Texas Tech
University is adjacent to the SWiFT site and was previously used to characterize the site (Kelley and Ennis, 2016).
For this experiment, we imagine testing over the months of September, October, and November during the hours of 9 AM
to 5 PM for five days a week (considering working hours for site operators). This filtering of times reflects the current require-
ments for attended operation at the SWiFT site, but, for experiments with unattended operation, then the full 24 hr/day met
data set would be used. This also points to the important distinction between the length of the measurement time and the length
of the experiment mentioned previously. In this virtual experiment, the experiment is imagined to last 3 months (2,184 hours),
but the measurement time when the turbine is being operated (and the length of time represented by the met data set) totals
only 480 hours spread over 3 months. As will be seen in the results, this has significant implications for the total time required
to meet convergence and significance criteria as an experiment that is not operating the turbines and recording data at all times
is missing at least some portion of the possible data collection. Finally, if this experiment were real, then this data should have
also been filtered to remove times when either turbine would have been in the wake of the other. Given the dominant wind





direction and to ensure there was enough data for this demonstration, however, this additional step was not taken.

Data from each met tower were filtered for these months and hours over multiple years and binned in 10-minute intervals. As inputs, TurbSim requires the mean hub height wind speed, turbulence intensity, and shear exponent, so these were calculated for each bin (Jonkman and Buhl, 2006). Note that other inflow parameters, e.g., density, do change over time and can significantly effect the results for some QoIs and may need to included. For simplicity, density was assumed to be constant in this demonstration.

The turbulence intensity was calculated as

$$TI_{10} = \frac{\sigma_{10}}{U_{\infty,10}}, \tag{5}$$

where $\sigma_{10}$ is the standard deviation of the hub height freestream wind speed in the 10-minute bin and $U_{\infty,10}$ is the average freestream wind speed in the 10-minute bin. For these simulations, the ScaleIEC parameter is turned on to ensure that the desired turbulence level is achieved. Liew and Larsen (2022), however, note that a similar scaling parameter in the aeroelastic code, HAWC2, causes a non-physical increase in energy at higher frequencies, so some caution may be necessary when interpreting results.

The shear exponent, $\alpha$, was calculated by fitting a power law between the wind speeds at two heights of the met tower such that

$$\alpha = \left( \ln\left(U_{45}\right) - \ln\left(U_{31}\right) \right) / \left( \ln\left(45\right) - \ln\left(31\right) \right), \tag{6}$$

where each $U$ is at a different height across the rotor plane. The shear exponent was then averaged for each 10-minute bin.

Since only the 10-minute statistics are needed, it is not necessary to apply quality control to the time series. Here, we used the site characterization data to set minimum and maximum allowable values for each parameter. Any bins with a parameter outside the allowable bounds were discarded. Any remaining errors in the inflow data are inherently propagated into the results and are not separately considered. If these errors are random, they will not alter the results as only a representation of the distributions of conditions is needed. Significant biases in the inflow measurements could, however, lead to erroneous results.

To represent the intended experiment, we use 2,520 10-minute bins randomly selected over the duration of the experiment. After filtering, 4,228 acceptable 10-minute samples remained from METa1 and only 1,443 remained from METb1. To get to 2,520 samples from each, samples from METa1 were randomly downsampled and samples from METb1 were randomly upsampled with replacement. Inflows with the same inputs may still produce different results because they will use a different seed in TurbSim. It should be noted that upsampling with replacement means that not every simulation is unique in the mean, which could bias our results. One way to assess this potential is to look at the distribution of good samples across months and hours to determine if there is adequate representation of the full time period, which is what is shown in Fig. 2. For each month and met tower, the number of samples for each day and hour are shown with a black line marking 100% data availability over all 12 weeks. The reason this is crossed in many plots is that data are taken from multiple years, so a particular day, for example, can have more than 48 samples in it (8 hr × 6 10-min bins per hr), and a particular hour can have more than 180 or 186 samples in it, depending on the month (30 or 31 days × 6 10-min bins per hr per day). Most days and all hours are undersampled, and





**Figure 2.** Histograms of the number of samples in each day and each working hour of each month for each met tower.

there are significant gaps especially in the METb1 data. While the hours are undersampled, they are at least fairly uniformly sampled, which should prevent any biases in hourly representation. The days, however, are very unevenly sampled for some

months, which could introduce biases, though day-to-day variations should be much smaller than hour-to-hour variations.

  These gaps could be filled in by interpolation based on the distributions of inflow parameters within the time period, but, without looking deeper into what conditions are and are not represented in the available data, it can be difficult to judge the effects of this undersampling. To some extent this can be seen in Fig. 3, which shows that the spread of turbulence and shear appears to be well-represented in all wind speeds. As this is just a demonstration of the method, we will proceed with an

acknowledgment that certain conditions may not be present in this data. When there are no significant gaps and the acceptable data provide a sufficient number of samples across the entire time period, upsampling with replacement should provide an



accurate representation of the statistical weight of each condition without biasing results. The end result of this procedure is a set of 10-minute statistics from each met tower that will each be used to generate one TurbSim inflow for one OpenFAST simulation, which will then represent one 10-minute sample of field data.

The resulting distributions of conditions from each met tower can be seen in Fig. 3. Data from METb1 appears sparser because many points are resampled, though it is clear that general trends in conditions are adequately represented in both data sets. Finally, a histogram showing the number of samples by 0.5 m s$^{-1}$ wind speed bins is shown in Fig. 4. This is useful to see both where there is more data in general and how the two data sets differ from each other in terms of the number of samples available in each wind speed bin.

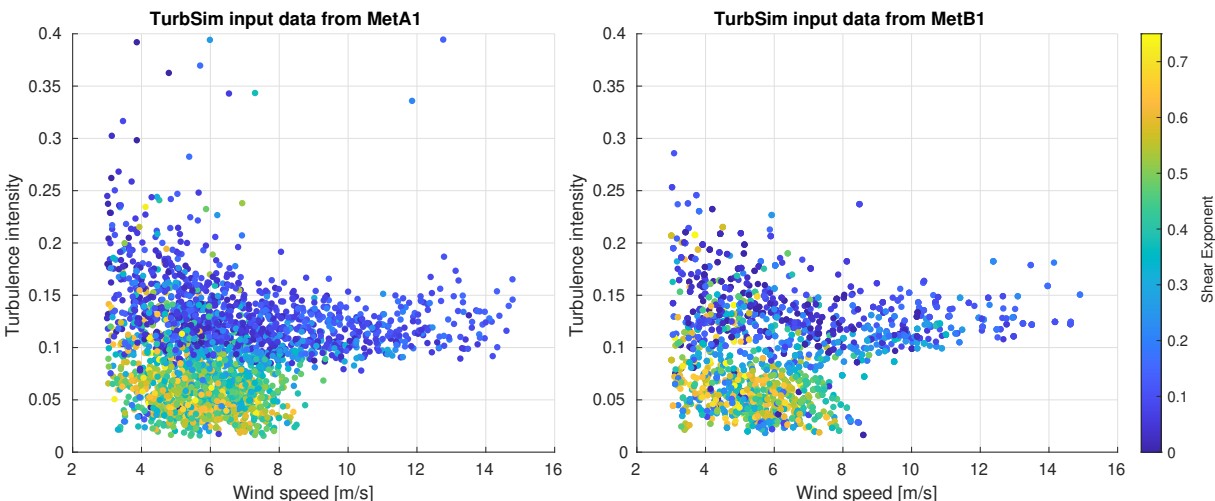

**Figure 3.** Scatter plots of wind speed by turbulence intensity with color showing the shear exponent. Each point represents one set of input parameters for a TurbSim inflow, though some points in the METb1 set are used more than once.






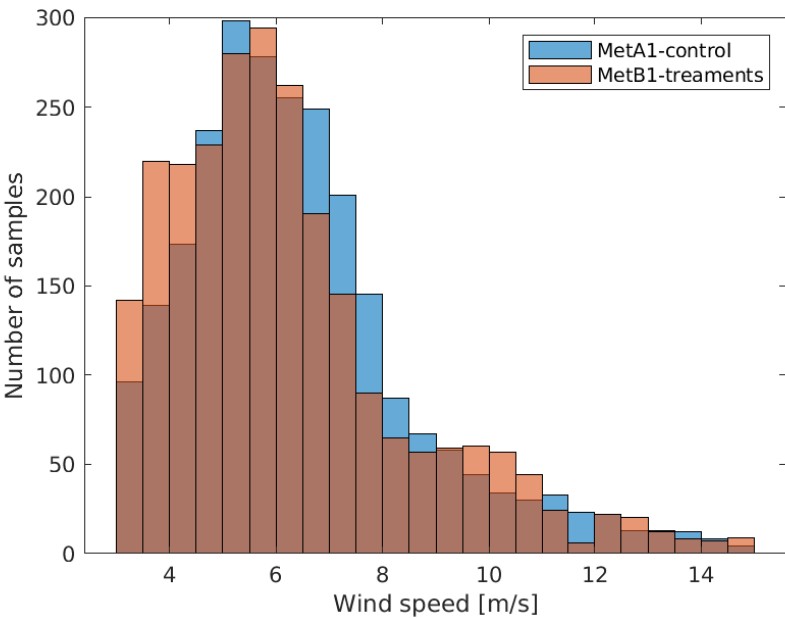

**Figure 4.** Histograms of each set of inflow conditions in 0.5 m s$^{-1}$ bins.

### 3.3 Case Study Simulations

All simulations are run using TurbSim generated inflows in OpenFAST. TurbSim uses the hub height wind speed, turbulence intensity, and shear exponent to numerically simulate time series of three-component wind speed vectors at points on a two-dimensional grid (Jonkman and Buhl, 2006). OpenFAST is a wind turbine simulator and provides modules such as InflowWind to accommodate TurbSim inflows, AeroDyn to calculate aerodynamics using a blade element momentum theory, ElastoDyn to calculate structural responses, and ServoDyn to calculate drivetrain and actuator responses (NREL, 2023). ROSCO is incorporated into ServoDyn to control the turbine relative to the unsteady, turbulent inflow. Using Sandia National Labs' high performance computing resources, each simulation is run on one node in approximately real time. In all, the control rotor is run with 2,520 different inflows from the METa1 data set to represent the control and each tip extension is run with 2,520 inflows from the METb1 data set to represent five different treatments.

### 3.4 Results from the Case Study

Before proceeding to results from the simulations, some observations can be made based on the inflow inputs. In Fig. 3, it is evident that there is much more variability in the inflow in terms of turbulence intensity and shear exponent at low wind speeds. We should anticipate that results at low wind speeds will converge more slowly and have higher random errors due to this. At high wind speeds, we see that there are generally fewer samples, which may make convergence impossible and may also not





support a robust bootstrap analysis. It is likely that this inflow data set will be inconclusive for results at the turbine's rated condition. Finally, Fig. 4 shows a clear mismatch in the number of available samples from each inflow data set at some wind speeds and this will have to be addressed carefully in the analysis.

In the proceeding results, we will consider power, thrust, flap root bending moment, and edge root bending moment. All

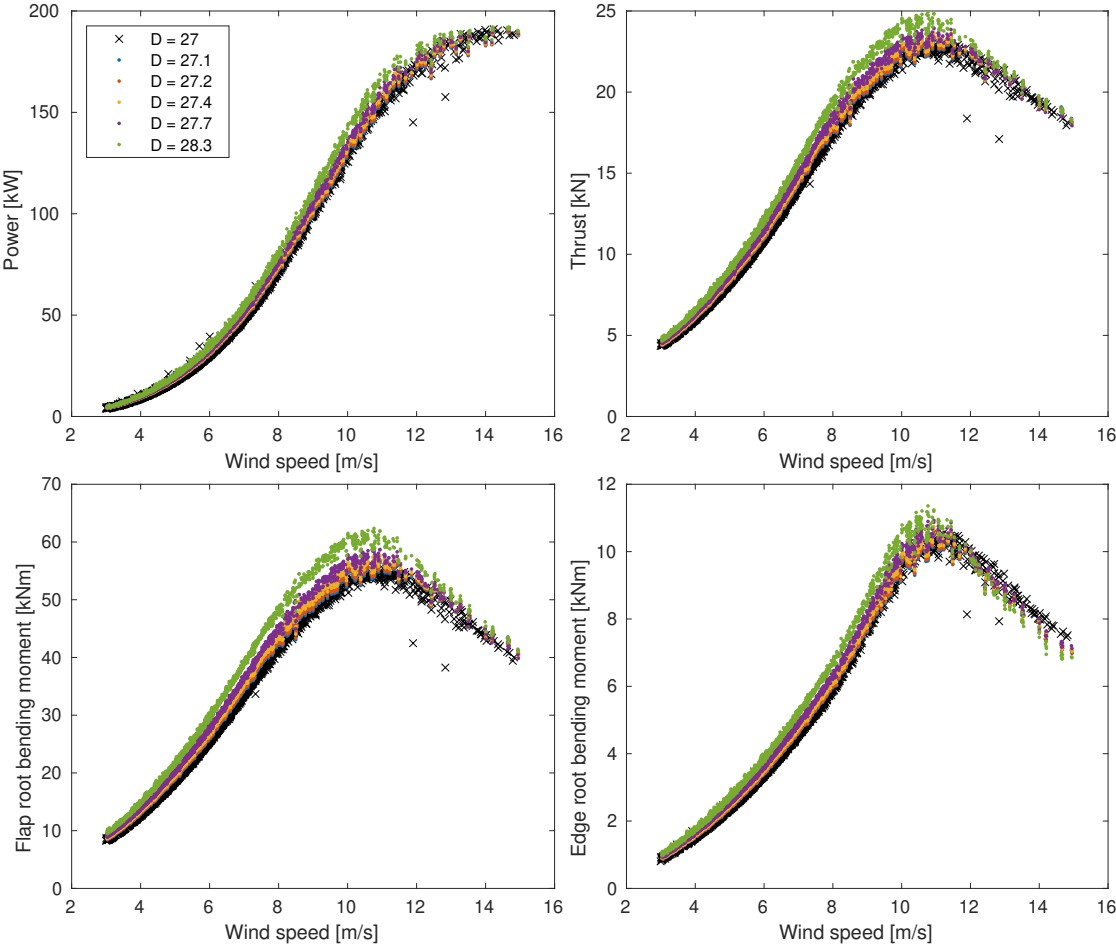

**Figure 5.** Raw output data from all simulations for each of the QoI to be considered.

QoIs are averaged from the last 10 minutes of each 700 s simulation and binned in 0.5 m s$^{-1}$ wind speed bins for the uncertainty analyses. Figure 5 shows the raw results from all simulations for our QoIs. Note that results from the tip extensions are expected to and do "stack" as they share an inflow data set while results for the control rotor stand out including some apparent outliers presumably as a result of its more complete inflow data set. Figure 6 shows the tip speed ratio (TSR) and the blade pitch





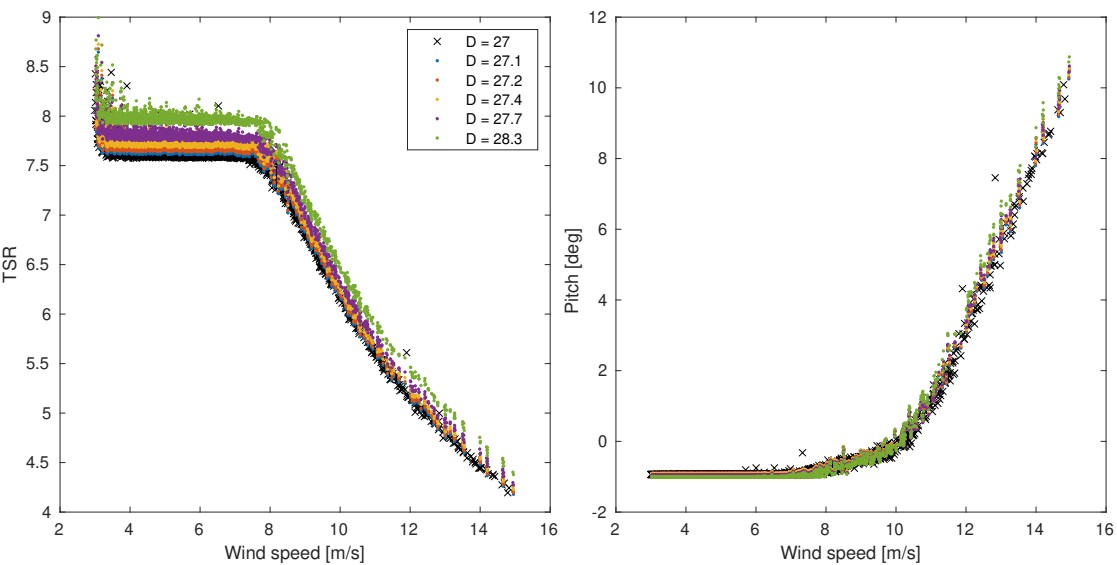

**Figure 6.** Raw output data from of the tip speed ratio (TSR) and blade pitch. Note the change in controls at 8 m s$^{-1}$ and at about 11 m s$^{-1}$

from every simulation to indicate when the control actions as a function of wind speed, which will be important to consider in
interpreting the data.

Figure 7 shows the standard deviation of each bin for each QoI normalized by its average (i.e., the relative standard deviation, or RSD) and reinforces the predictions made by looking at the inflow parameters. We see that, in general, the RSD is higher at lower wind speeds due to the higher variability of inflows at low wind speeds. It is interesting to note that this does not affect all QoIs equally, though, as the RSDs for power and edge root bending moment are approximately double that of thrust and
flap root bending moment. Furthermore, the spikes at high wind speeds are likely additional indicators of inadequate numbers of sample points such that the distributions are not well represented.

Figure 8 shows the percent relative random error (i.e., the random error of the QoI as a percent of its ensemble mean) using the bootstrap analysis (with no minimum sample number applied yet) for each QoI in each wind speed bin for all rotors. The differences between the control and treatments are primarily a result of the different numbers of samples in each bin from their
different inflow data sets. As predicted based on the inflow data, the uncertainties at lower wind speeds are the highest due to the higher variation in conditions when only binning on wind speed. We also see how the observations made regarding Fig. 7 propagated through to higher errors in power and edge root bending moment at low wind speeds than for thrust and flap root bending moment. Again, results in the highest five wind speed bins should be interpreted cautiously as there are possibly too few samples in these bins to make conclusions. Finally, the increase at 8 m s$^{-1}$ is due to the control actions as this marks the
beginning of region 2.5 for all rotors. As the raw data in Fig. 5 are smooth across this transition, it is a result of the binning that additional variance is highlighted due to the change in controls. This is also evident to a lesser extent in Fig. 7. This highlights



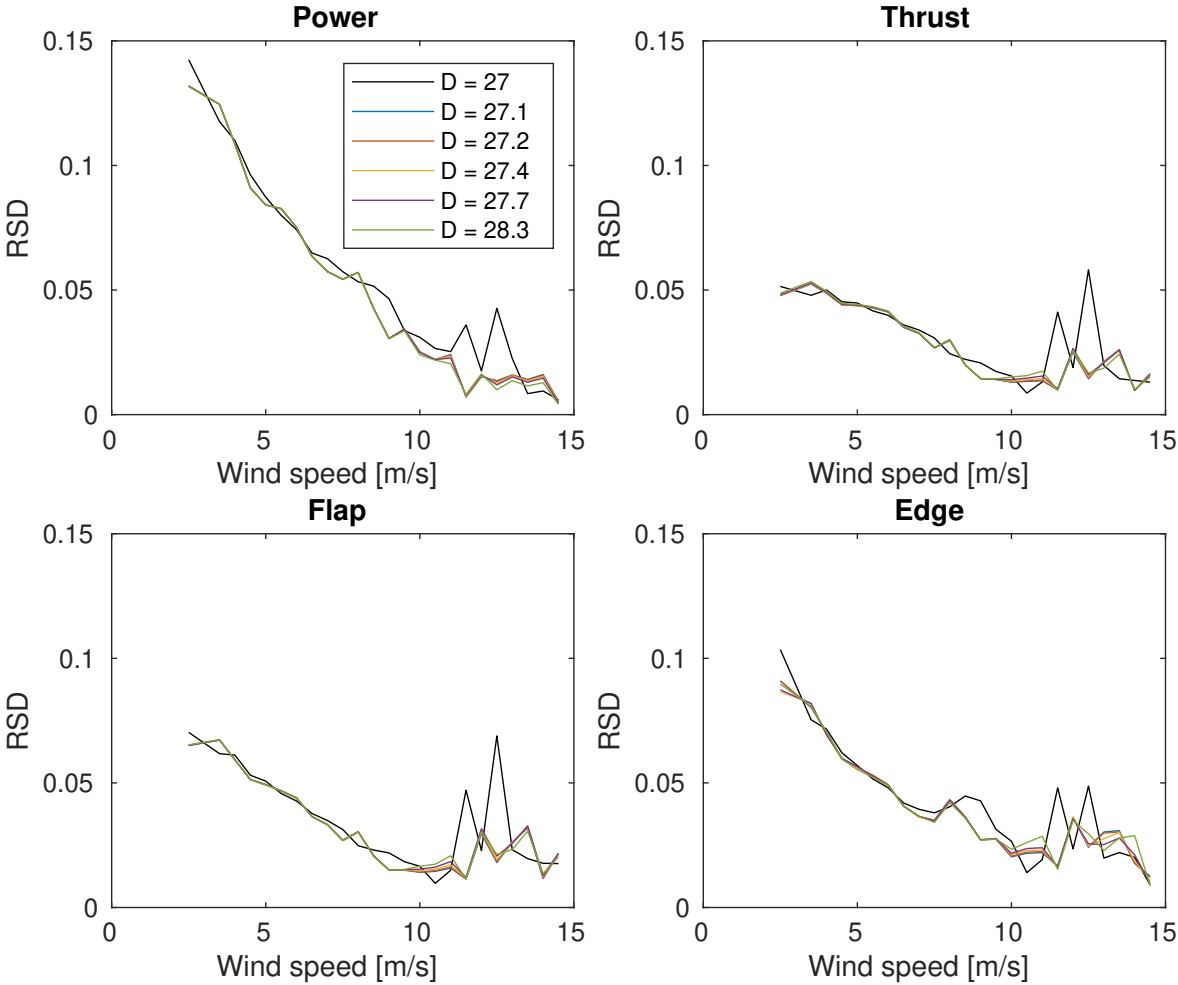

**Figure 7.** The standard deviation of each QoI in each wind speed bin normalized by the average of the same, or relative standard deviation (RSD).

another important consideration in wind turbine field experiments, namely that conditions that trigger a change in control may produce increased variability in some QoIs and therefore require additional measurements to converge and produce significant results after binning.

Recall that the real goal of this virtual experiment is to determine the measurement and experiment durations required for converged and significant differences, though care must be taken here on several points. First, the METa1 and METb1 data sets do not have the same number of samples in each wind speed bin. To calculate the running error in differences between the control and treatments, the running mean of the control QoI is subtracted from the running mean of each treatment QoI for





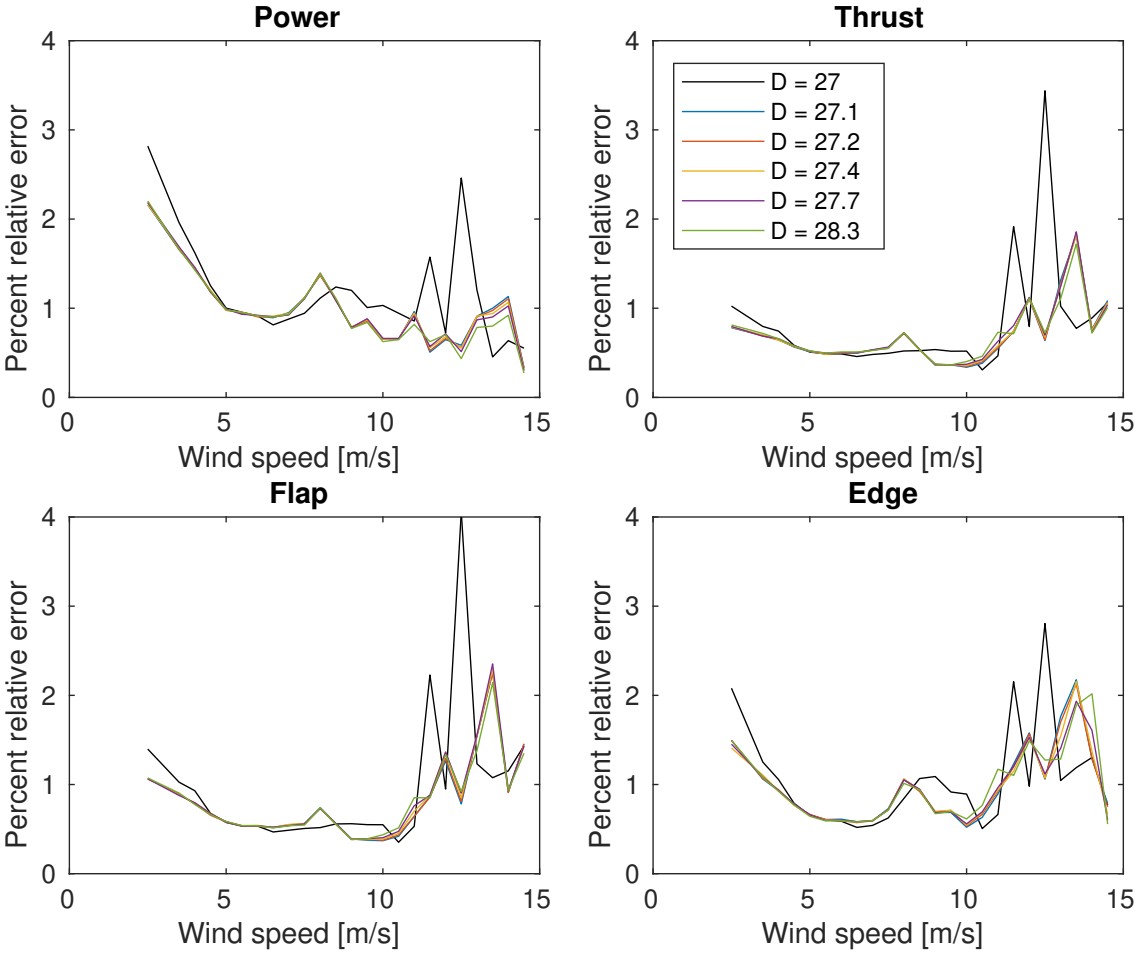

**Figure 8.** The percent relative error (i.e., the error of the QoI as a percent of its ensemble mean) of each QoI calculated for each wind speed bin using all available samples.

each wind speed bin as long as samples remain in both data sets. When one reaches its last sample (i.e., the ensemble mean

for that bin), that value is held and the subtraction proceeds until the other has used all of its samples. In a similar manner, the individual errors associated with the control and treatments are added in quadrature for a given pair to produce a running uncertainty interval on the running difference. Having now defined the running difference and uncertainty intervals for each combination of the control and a treatment, the data are easily filtered to find the sample at which a significant difference is achieved (i.e., zero is no longer within the uncertainty interval) and remains true. Herein, we have arbitrarily chosen to use a

95% confidence interval. The results of this step are shown in Fig. 9 just for the differences in power.

      Mathematically, the data in a bin may become and remain significant with only one sample, which suggests an additional





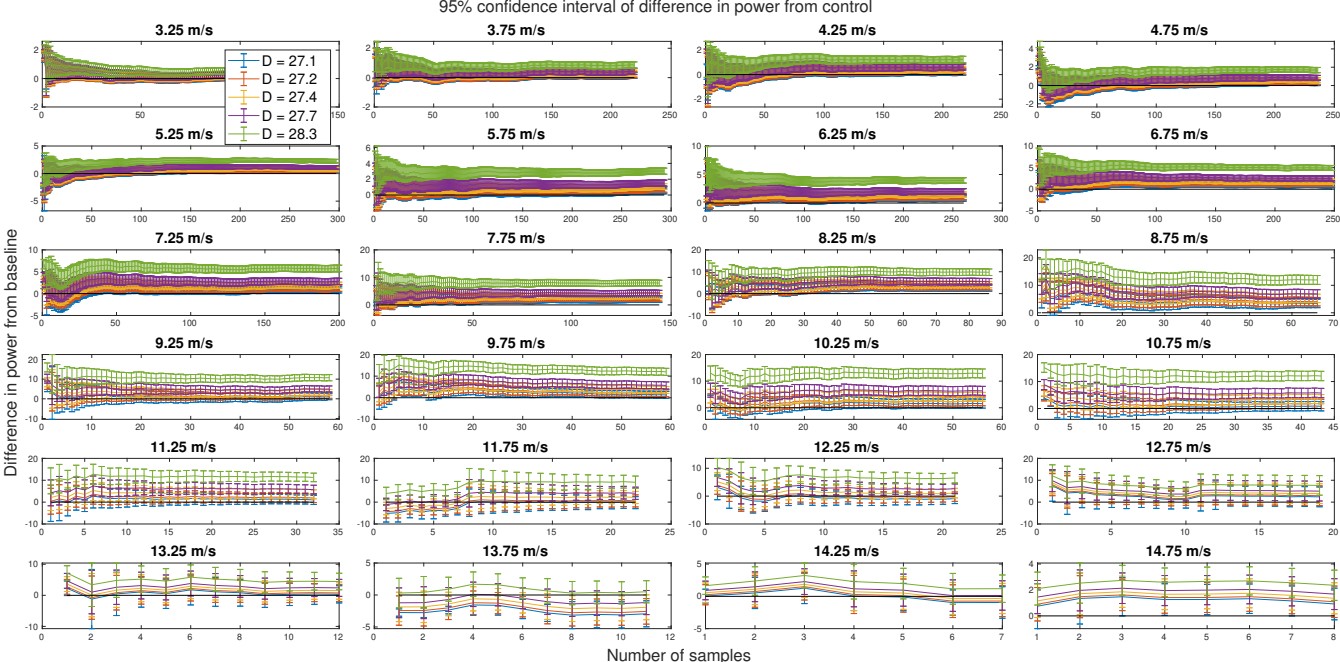

**Figure 9.** Error bars on the running difference in power for each treatment rotor from the control in each wind speed bin. The black line marks zero to more clearly tell when differences are significant.

need for a convergence criterion, which is separately implemented. Here, it is required that the running mean of the difference in a QoI between the control and a treatment be less than and remain less than 2% of the ensemble mean in each bin. Because all bins will, by definition, converge to zero difference between the running and ensemble means, this standard is further

required for two consecutive samples not including the last sample (when this difference is always zero). This has the effect of putting a restriction on the rate of convergence. The standard for convergence is somewhat arbitrary. Here, 2% was chosen as it is approximately the average percent relative error (see Fig. 8,) so further convergence likely remains within uncertainty. An example of the convergence of the difference in power is plotted in Fig. 10. Having marked at which sample number (the samples remain in order of time throughout) the data are converged and at which a significant difference is achieved, the greater

of those two is taken as the minimum required number of samples in a wind speed bin for a given QoI for a given treatment.

    Next, the data are filtered to ensure that each bin has a minimum number of samples for a robust bootstrap analysis as discussed earlier. Given that this data set is somewhat bimodal both in its inputs (the variability in inflow as shown in Fig. 3 is much higher below about 9 m s$^{-1}$) and in its outputs (the turbine controller switches at 8 m s$^{-1}$ to region 2.5 controls), we use the 9 m s$^{-1}$ bin as a dividing line between two minima. For all bins below 9 m s$^{-1}$, we require at least 25 samples due to the

high variance in inflow conditions. For bins higher than and including 9 m s$^{-1}$, we require only 8 samples as the variability is markedly reduced at high wind speeds (Jenkins and Quintana-Ascencio, 2020).

    The final step is to use the timestamps from the original met tower data to convert samples marked as having met all criteria





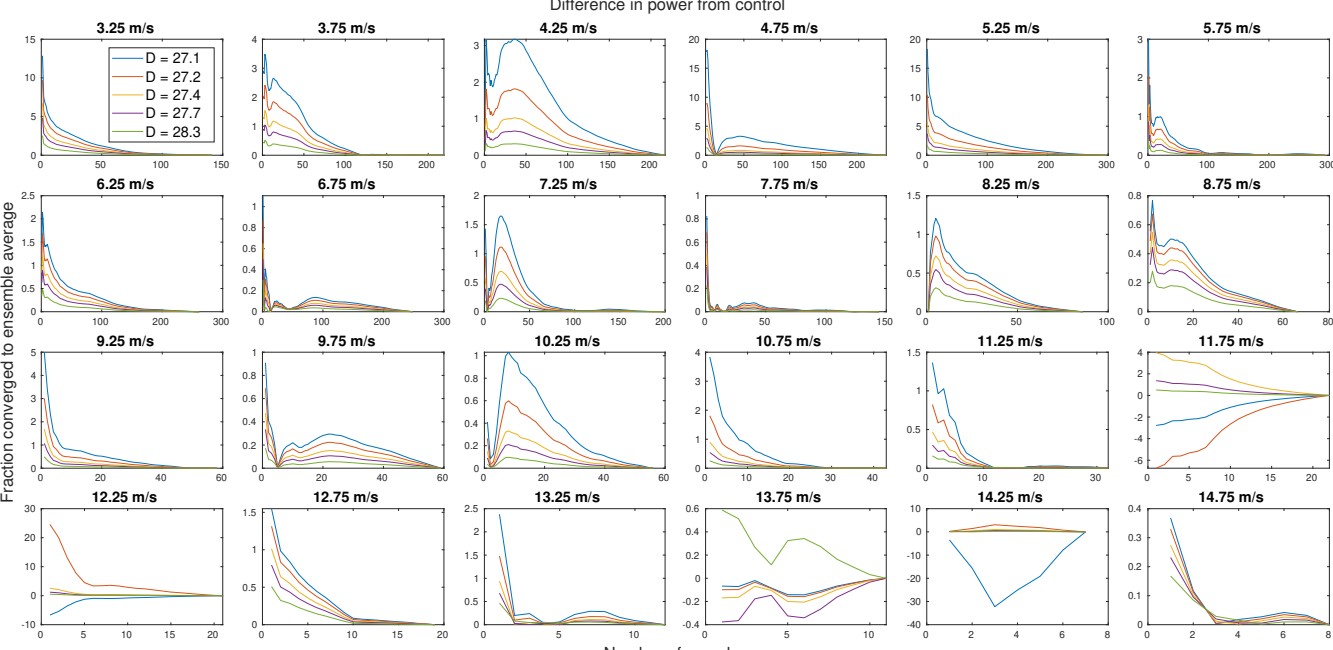

**Figure 10.** The convergence of the difference in power of each treatment rotor from the control for each wind speed bin.

into measurement and experiment durations required relative to the start of the experiment. Here, one final check is required to ensure accurate results. Because the inflow data are taken from multiple years and are in 10-minute bins, it is possible that

some samples are coincident when ignoring years (i.e., they have the same date and time). If not addressed, this would lead to undercounting of the durations based on timestamps. To prevent this, a final correction is made such that, if the time required to meet all criteria is less than the number of samples to meet all criteria times 10 minutes per sample, then the latter is taken as the time to meet all criteria.

Figures 11-14 show the final results of this virtual experiment. In all plots, a scale of time shows the number of experiment

(not measurement) hours required for each treatment to produce a significant and converged difference over the control and are binned by wind speed. The durations plotted in these figures are from the beginning of the experiment and include all time (i.e., not just the time during which measurements were being recorded) such that they provide the total duration required including time simply spent waiting for the conditions necessary to fill out a particular bin. In other words, in this virtual experiment, the turbines are only operated during weekday working hours, so approximately 80% of each week is not included in the

measurements, which requires more total days of operating to provide enough data to meet the criteria for convergence and significance in each bin. In this experiment, the maximum possible duration, including times we filtered out (e.g., outside of working hours), is about 2,184 hours. This highlights one of the main challenges of field experiments: we do not control the wind! Much, if not most, of the duration required to produce converged and significant results is, in essence, time spent waiting for the right conditions especially if you are not always operating and measuring.




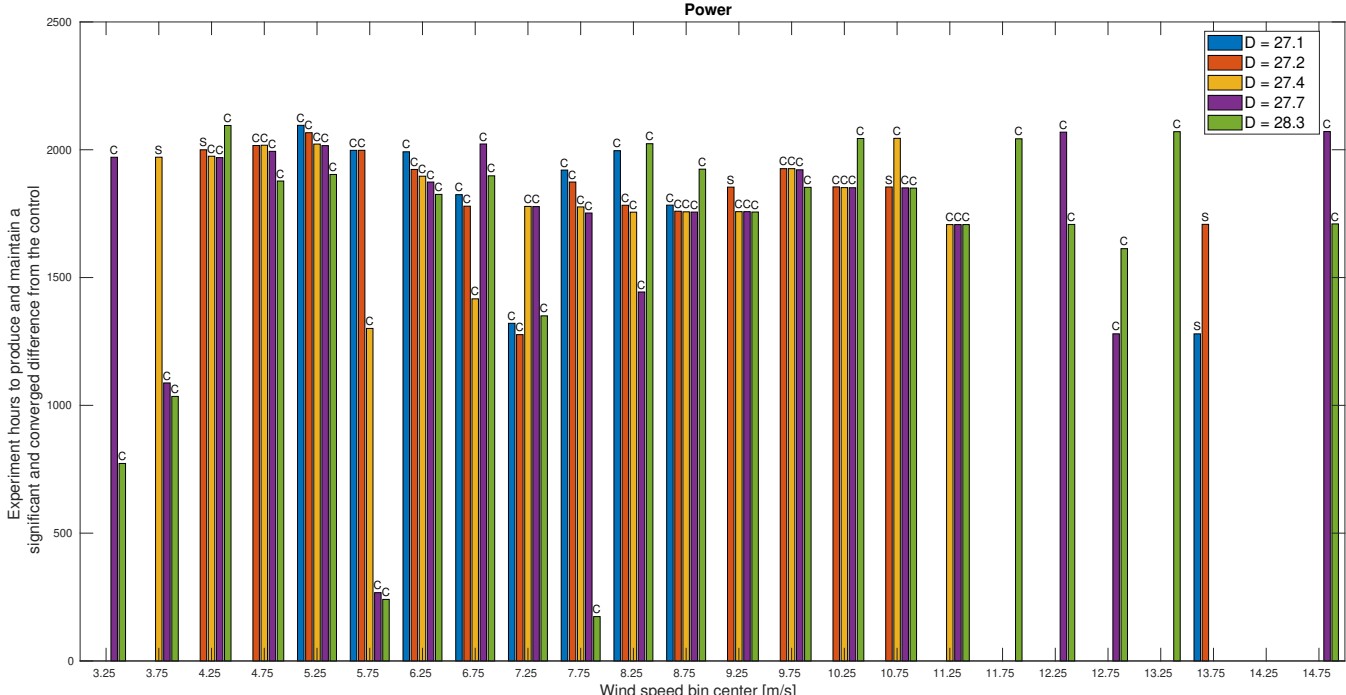

**Figure 11.** The minimum experiment duration required to produce a significant and converged difference in power between the control and treatments. Whether the minimum time was dictated by convergence (C) or significance (S) is indicated above each bar. Missing bars indicate that a significant and converged difference was not achieved within the simulated experiment time.

Some general trends are observable in all QoIs. First, we see that it is more likely for a treatment to pass all criteria in the middle wind speeds than at low wind speeds and especially at high wind speeds. At low wind speeds, though there may be many samples, the high variance of the inflow makes it more difficult for results to converge. At high wind speeds, however, there are two possible reasons that few rotors meet the criteria: There are simply are not many samples in these bins, and, in region 3, the differences in power are reduced so significance becomes more challenging. In these bins, this method is somewhat inconclusive as we cannot say how many more samples would be required to pass all criteria; we can only say that there were not enough for this analysis. Second, for almost all QoIs, rotors, and wind speed bins, it is convergence and not significance that dictates the minimum required time. In other words, the rate at which convergence is achieved is slower than the rate at which significance is achieved. In fact, it is almost exclusively the smallest three rotors for which significance ever dictates the minimum time. It is precisely because these rotors produce smaller differences that they converge before they become significant. Similarly, for most QoIs and bins, the largest rotor requires less time to meet all criteria. As convergence is primarily a function of inflow conditions, this can be attributed to the larger rotor producing larger differences and thereby reaching significant differences with fewer samples.

Across all QoIs, however, there are several wind speed bins that do not follow the pattern we might expect that, generally

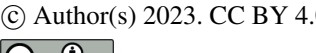



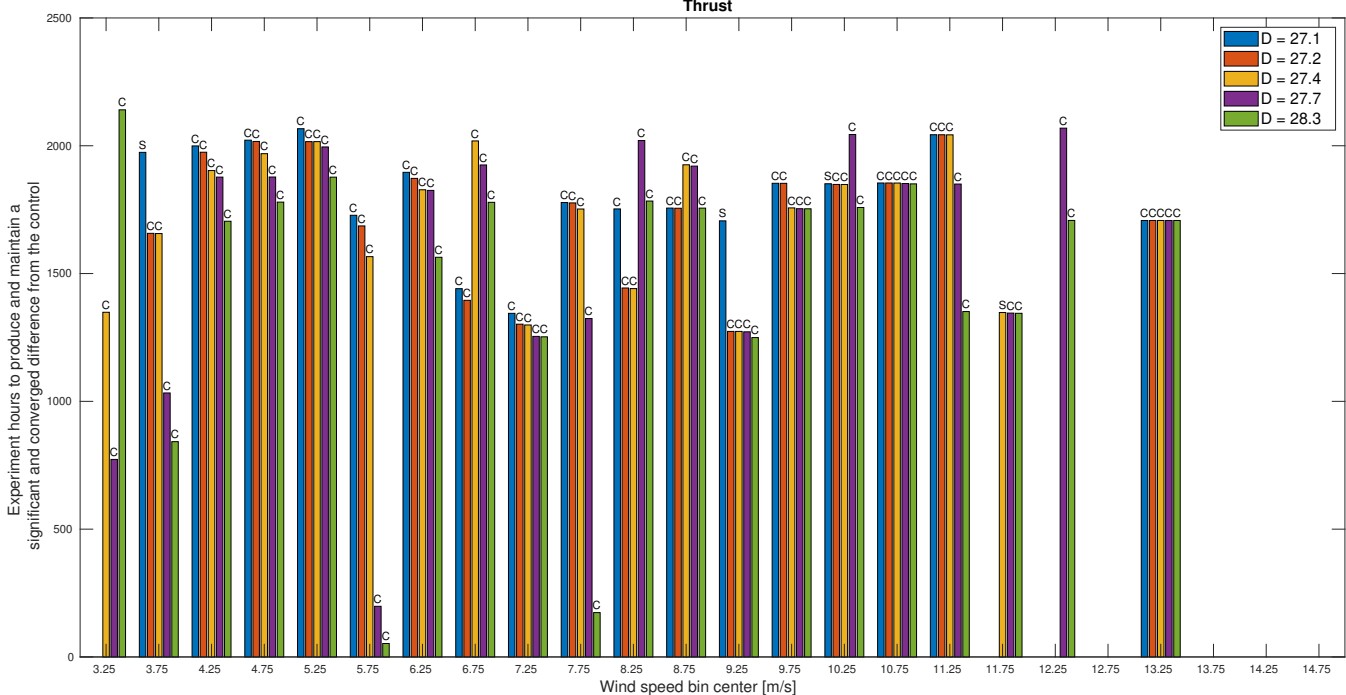

**Figure 12.** The minimum experiment duration required to produce a significant and converged difference in thrust between the control and treatments. Whether the minimum time was dictated by convergence (C) or significance (S) is indicated above each bar. Missing bars indicate that a significant and converged difference was not achieved within the simulated experiment time.

speaking, the larger rotors would produce larger differences from the baseline and so require shorter durations to measure.

Though the rotors were only designed based on an expected difference in power, it follows that we would expect proportional changes in thrust and flap and edge root bending moments. The one pattern that does emerge is which wind speed bins do not adhere to this expectation. Across these four QoIs, the bins centered on 6.75, 8.25, and 10.25 m s$^{-1}$ exhibit orders of decreasing time per rotor other than smallest rotor to largest, though not necessarily the same order for different QoIs. Similarly, the bins centered on 3.25, 4.25, and 8.75 m s$^{-1}$ also do not adhere to this expectation for three of the four QoIs. These six wind speeds

correspond approximately to transitions in the turbine controls specifically cut-in, region 2.5, and region 3 (see Fig. 6) and also correspond to apparent increases in variance as seen in Fig. 8. Similarly, it is controller actions that cause only the smallest two rotors to meet criteria in the bin centered on 13.75 m s$^{-1}$. As shown in Fig. 9, the trend in differences in region 3 is generally toward zero, but in this bin, the two smallest rotors cross into a negative difference that becomes significant. This result is likely sensitive to the bin width.

A few specific results require further attention. First, in Fig. 11, we see that, for most rotors and most wind speeds, we can expect to need to run the experiment for over 1,500 hours when only measuring during weekday working hours to produce any significant and converged results and possibly over 2,000 hours for the smaller rotors. The result for thrust in Fig. 12 paints a





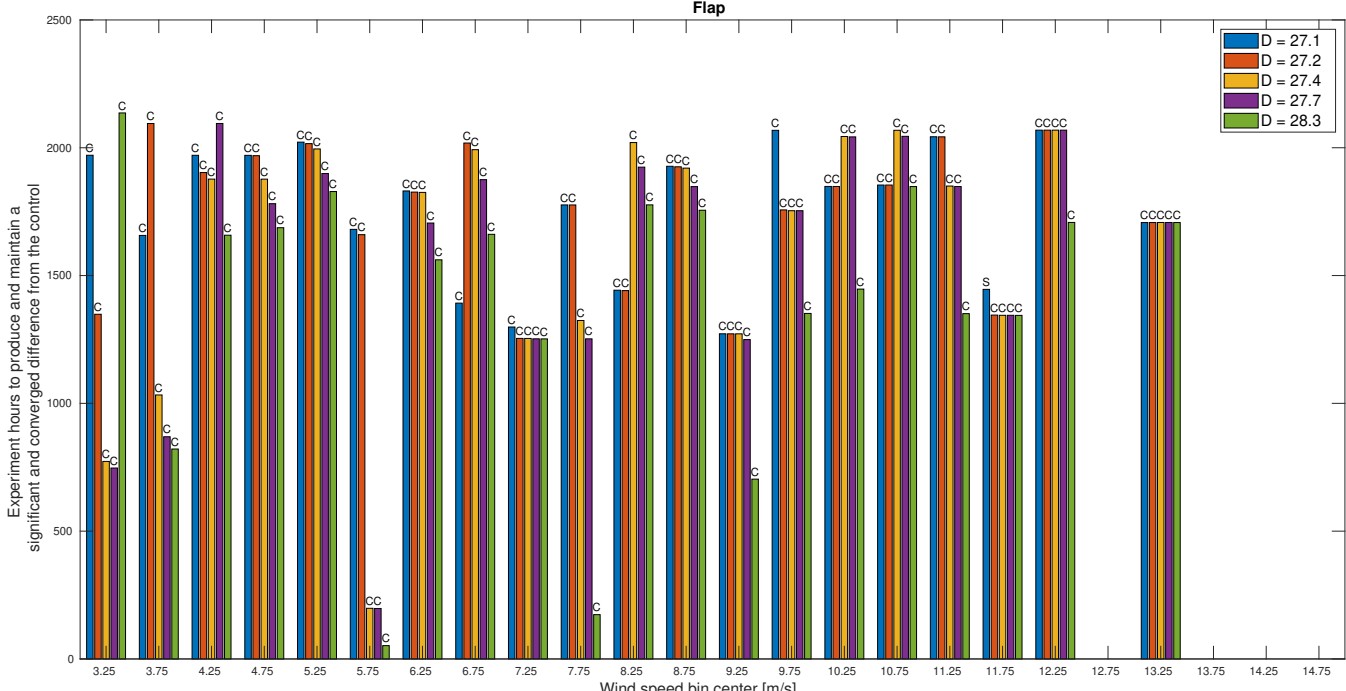

**Figure 13.** The minimum experiment duration required to produce a significant and converged difference in flap root bending moment between the control and treatments. Whether the minimum time was dictated by convergence (C) or significance (S) is indicated above each bar. Missing bars indicate that a significant and converged difference was not achieved within the simulated experiment time.

similar picture, though the time required for a smaller rotor is shorter than for power at most wind speeds and more rotors in more bins meet both criteria. It also appears that, in the case of thrust, there may be a more uniform minimum time across rotor
sizes as we see smaller differences in the times required per rotor especially at higher wind speeds.

In Fig. 13, it appears that flap root bending moment requires similar measurement times as the other QoIs and also appears to have minimum thresholds in the middle and higher wind speeds such that larger rotors require the same amount of time as smaller ones. Finally, in Fig. 14, it is clear that the edge root bending moment is a challenging measurement as many entire bins as well as the smaller rotors in many other bins, in particular higher wind speeds, do not meet the criteria within the
total simulated experiment time. A particular challenge of the edge root bending moment is the reversal in trend by rotor size at around 12 m s$^{-1}$ as seen in Fig. 5, which will reduce differences in this and perhaps neighboring bins. For both flap and edge moments, we would frequently be interested in capturing the peak loading condition at around 11 m s$^{-1}$ (see Fig. 5). Despite exhibiting the largest differences among rotors at this peak, this would be a challenging measurement to capture in this experiment simply because these wind speeds do not occur as often as lower ones.
To further emphasize the difference in the minimum experiment duration and the minimum measurement duration, Fig. 15 shows the minimum measurement time for each treatment rotor to reach a significant and converged difference in power from





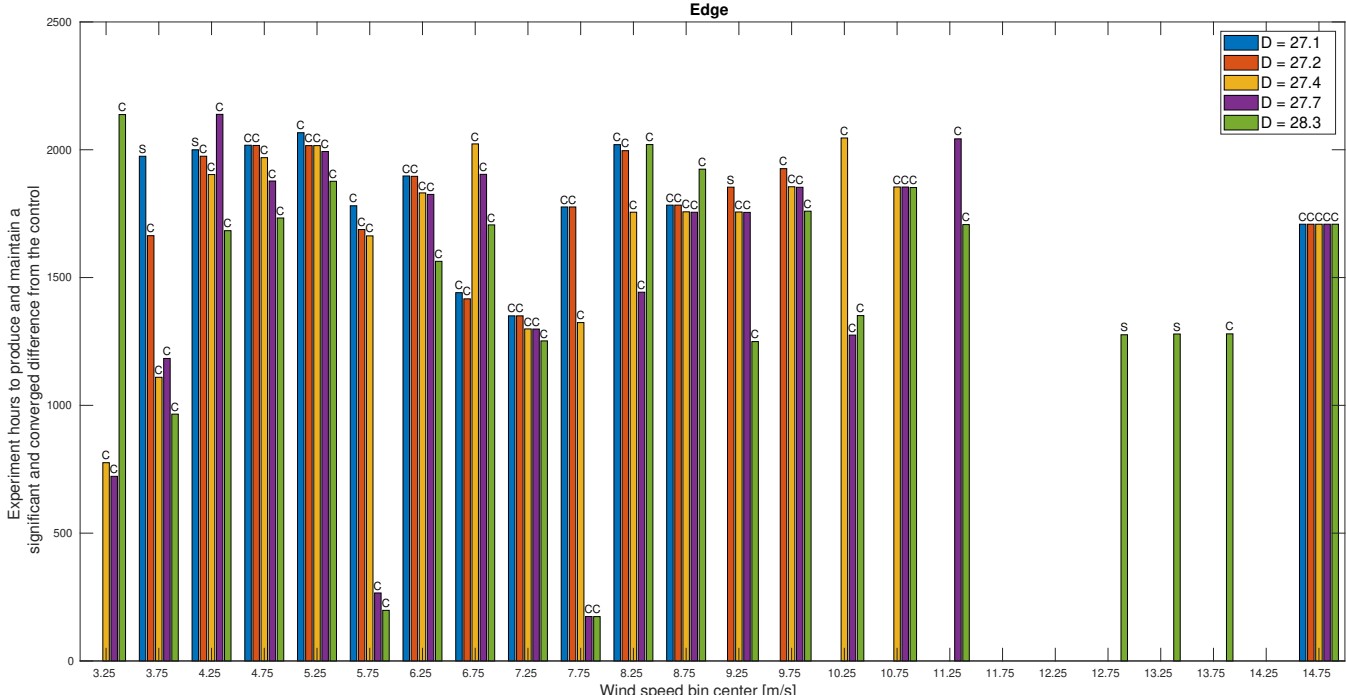

**Figure 14.** The minimum experiment duration required to produce a significant and converged difference in edge root bending moment between the control and treatments. Whether the minimum time was dictated by convergence (C) or significance (S) is indicated above each bar. Missing bars indicate that a significant and converged difference was not achieved within the simulated experiment time.

the control rotor by simply converting the number of samples required into the equivalent amount of time. Note the difference in scale from the experiment time. All bins that achieve a significant and converged difference from the control do so in fewer than 50 hours of measurements. This time, however, is spread across the entire experiment duration in some cases. The general trend

remains the same, namely that the smaller rotors require more time (which is equivalent to number of samples here). However, plotting the measurement as opposed to the experiment duration reverses the trend relative to the distribution of wind speeds (Fig. 4). For experiment duration, the more frequent wind speeds typically increased the likelihood of reaching a converged and significant result and thereby reduced the time required, whereas, for measurement duration, we see the opposite. This comes from the interplay of the variance exhibited in the inflow as a function of wind speed (Fig. 3) and the frequency of or number of

samples in a wind speed bin (Fig. 4). The variance in a bin has the same effect on both experiment and measurement duration such that higher variance increases the challenge of achieving convergence and significance. The frequency of or number of samples in a wind speed bin requires a more subtle interpretation. For both experiment and measurement duration, the number of samples acts as a threshold to whether or not there is sufficient data for convergence and significance. For the experiment duration, however, it must also be thought of as a frequency of occurrence such that more frequent conditions are more likely to

reach convergence and significance within the experiment duration, whereas, for measurement duration, when the samples are



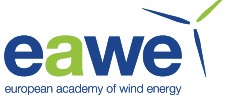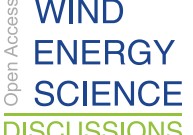

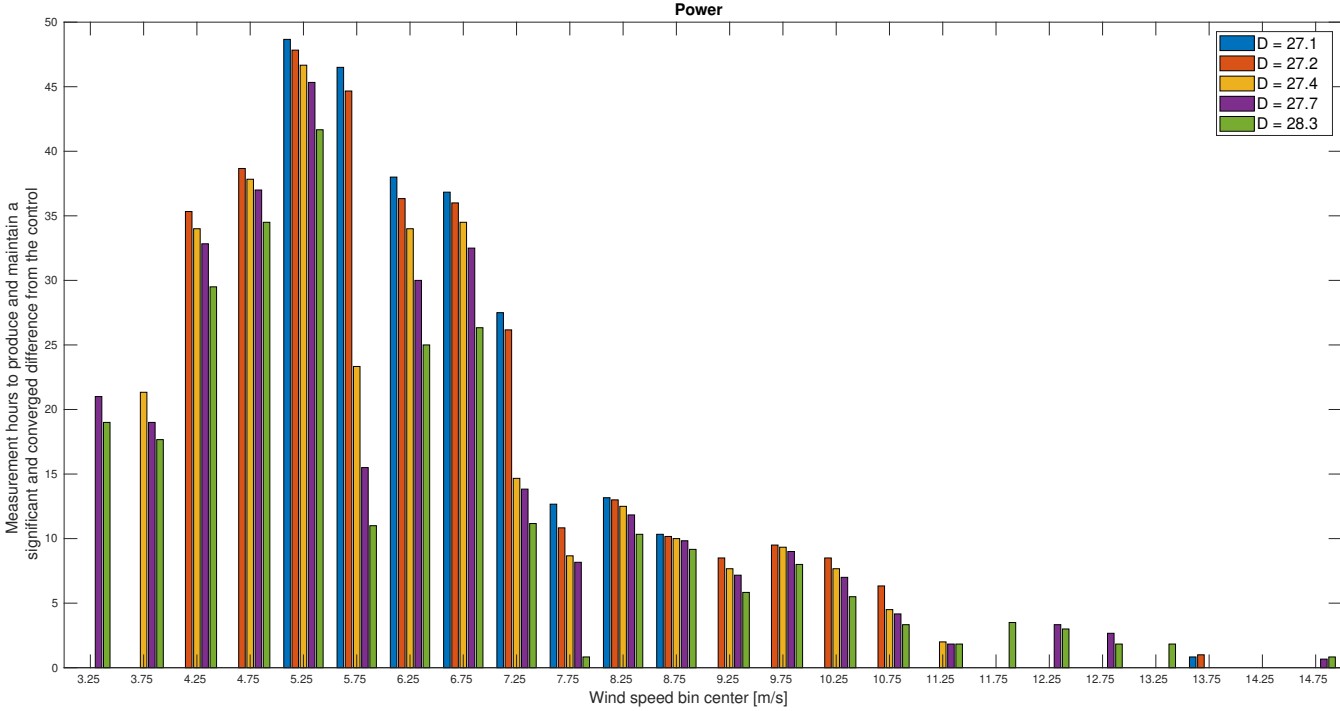

**Figure 15.** The minimum measurement duration required to produce a significant and converged difference in power between the control and treatments.

collected is irrelevant and it only matters that there are enough of them. Considering the variance and the number of samples in a bin, we see that the lowest wind speeds have much higher variance in TI and shear and they also have relatively fewer samples. That is reflected in Fig. 15 in that the smaller rotors do not have sufficient measurement time to reach convergence and significance with such high variance at the lowest wind speeds. Then, as wind speed increases, variance decreases, and

the number of samples increases, additional rotors meet the criteria until the measurement duration required peaks at around 5.5 m s$^{-1}$. As wind speed increases further, the variance in conditions slowly decreases and so does the measurement duration required. At the highest wind speeds, there are too few samples over the duration of the experiment for most rotors to meet the criteria.

## 3.5 Discussion of the Case Study

As presented in this virtual experiment, this method would allow the experimenter to plan an experiment with an expected difference in power from the control and to know the minimum measurement and experiment durations required to ensure significant and converged results within the standards used, namely a 95% confidence interval and convergence within 2% of the ensemble mean. For wind speed bins that did not have enough data to meet these criteria, additional time could be simulated to find the minimum. It is worth noting, however, that another approach could easily be taken within the same method. As it is



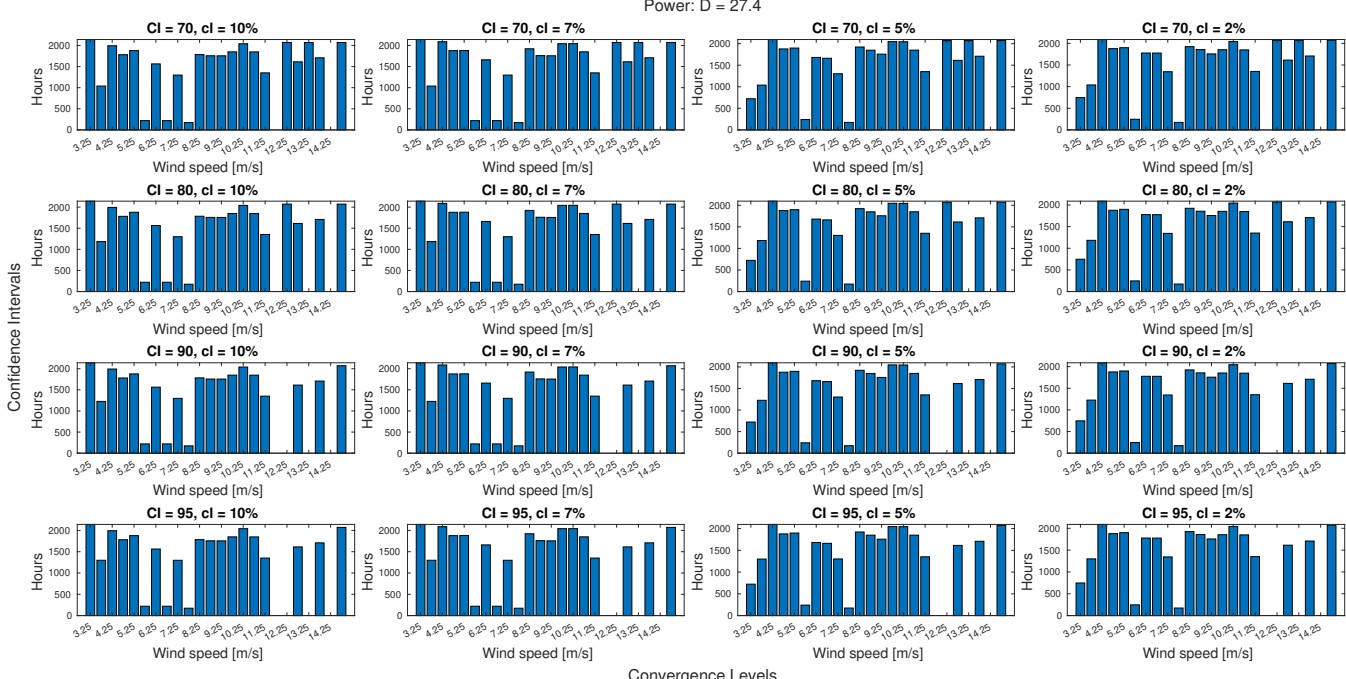

**Figure 16.** An array of plots showing how the experiment duration required to measure a significant and converged difference in power for one particular treatment rotor changes as the confidence interval (CI) and convergence level (cl) change.

frequently the case that time, funding, and/or equipment are restricted when planning an experiment, an experimenter may be interested to know what levels of significance and convergence could be achieved within a fixed experiment duration. In this case, the post-processing steps could add confidence interval and convergence level as parameters over which to view the results within a fixed duration and thereby determine what could be achieved in this duration as opposed to the duration required to achieve given standards. An example of this can be seen in Fig. 16, which shows how the minimum experiment duration

required changes as a function of confidence interval and convergence level. Since many bins are near the total experiment duration simulated with the most lenient standards, a stricter confidence interval can cause some of these bins to no longer meet the criteria (e.g., the bin centered on 12.25 m s$^{-1}$). A stricter convergence level, however, tends to just require a little more time. Many of the changes across confidence interval and/or convergence level are fairly small and difficult to discern, perhaps the addition of just a few additional samples. Figure 17 makes this clearer by showing just one wind speed and rotor across a

larger set of confidence intervals and convergence levels.

     Finally, it should be emphasized that some of the trends observed in this virtual experiment may not be found in other experiments. The specific trends identified are possibly, and even likely, specific to the experiment. One additional, though unconfirmed, possibility of this method is, however, the ability to simulate a surrogate for a more complex experiment. For example, this work was inspired by the additively manufactured, system integrated tip (AMSIT) project in which the tips of

a set of blades will be replaced by additively manufactured tips with a winglet and aerodynamic surface texturing as opposed




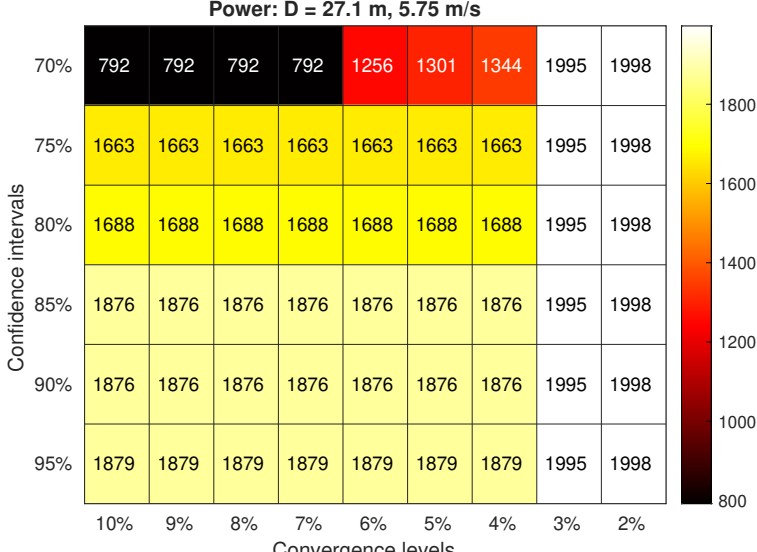

**Figure 17.** A heatmap showing the experiment duration required to reach a significant difference in power for one particular treatment rotor and one wind speed as a function of the confidence interval and convergence level.

to the straight and smooth original tips (Maniaci et al., 2023). There was insufficient data available in literature to say with confidence if OpenFAST could simulate the extreme curvature of the winglet or if we could represent the effects of aerodynamic surface texturing through alterations to the airfoil polars. In lieu of these changes, we opted to simply increase the area of the existing rotor through tip extensions to parameterize results by the expected change in power. It remains to be seen if

this yields a comparable result to having simulated the proposed AMSIT rotor. Similarly, blades or controls could be altered in other ways to influence other QoIs or inflows prescribed to suss out the effects of specific conditions.

## 4 Conclusions

A method to aid in predicting and potentially reducing experiment uncertainties, especially in the case of field experiments,

has been presented. The method requires inflow data either in the form of historical data from the experiment site or probabilistic distributions and a simulation method that balances fidelity with computational time. By running many simulations that represent the proposed experiment and performing uncertainty analyses on the results, an experimenter can better estimate the measurement duration required to produce converged and significant results and the experiment duration required to achieve this. Additionally, the simulated data can be used to try different analysis methods such as binning procedures or turbine control

switching to further estimate their effects on uncertainty and required durations.

To demonstrate this method, an experiment was imagined in which five tip extensions were compared to a control rotor in measurements simulated over a three month period. Power, thrust, and flap and edge root bending moments were compared.



Even before looking at the simulation outputs, general trends were predicted based on the experiment set up and inflow conditions. Namely, as predicted, the larger rotors generally required less data, and so typically less time, to produce significant results because they produce larger differences that can tolerate larger uncertainties. From the inflow conditions, it was correctly predicted that having more data in a bin would allow for QoIs to converge and reach significant differences in less time. Additionally, we correctly predicted that the high variance in conditions at low wind speeds and the lower sample counts at high wind speeds would make it more challenging to produce converged and significant results at those wind speeds.

In analyzing the final data produced from the simulations, we found that all QoIs investigated generally required similar experiment durations, though the edge root bending moment was especially challenging to capture at high wind speeds. The experiment duration required for the majority of results was dictated by convergence, not significance, except in the case of the smallest rotors for which significance was the more challenging criteria. It was observed that the wind speeds at which the turbine controls change their operation can be especially challenging as this can lead to increased variance in the QoI after binning. It is possible that non-uniform binning would improve results around these wind speeds by widening some wind speed bins. Finally, the minimum required experiment duration was compared to the minimum required measurement duration to emphasize that, when measurements are not being continuously recorded, a significant portion of the time required to achieve significant and converged results is essentially time spent waiting for the necessary conditions. Discontinuous measurements increase the experiment time required to have enough samples in each bin to ensure significance and convergence are achieved.

In closing, we emphasize again that this method is highly adaptable. While we focused on the challenges of field experiments, this could also be used for a suite of wind tunnel measurements or simulations. It is, in fact, generalizable beyond wind energy as long as the experimenter has a good understanding of how to simulate the experiment and the parameters that will have the greatest effects on the measurements.

*Author contributions.* DH was responsible for conceptualization, methodology, investigation, formal analysis, original draft preparation, and review and editing; ND was responsible for methodology, software, and review and editing; DM was responsible for supervision; and BC was responsible for funding acquisition, project management, supervision, and review and editing.

*Competing interests.* The authors declare that they have no conflict of interest.

*Disclaimer.* This article has been authored by an employee of National Technology & Engineering Solutions of Sandia, LLC under Contract No. DE-NA0003525 with the U.S. Department of Energy (DOE). The employee owns all right, title and interest in and to the article and is solely responsible for its contents. The United States Government retains and the publisher, by accepting the article for publication, acknowledges that the United States Government retains a non-exclusive, paid-up, irrevocable, world-wide license to publish or reproduce the published form of this article or allow others to do so, for United States Government purposes. The DOE will provide public access to



these results of federally sponsored research in accordance with the DOE Public Access Plan https://www.energy.gov/downloads/doe-public-access-plan.

*Acknowledgements.* This work was completed as part of the Additive-Manufactured, System-Integrated Tip (AMSIT) wind turbine blade
510   project, funded by the U.S. Department of Energy Advanced Manufacturing and Materials Technologies Office (AMMTO).



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
