# Peer review of "Method to predict the minimum measurement and experiment durations needed to achieve converged and significant results in a wind energy field experiment"

_Wind Energy Science, 2023_

## Referee Comment (RC1)

*Discussion of:*

**Predicting and reducing wind energy field experiment uncertainties with low-fidelity simulations**

09 December 2023

The paper presents a virtual experiment-based methodology for estimating the amount of uncertainty in wind turbine load and power performance measurements, subsequently providing possibilities to estimate how long a measurement campaign will need to be in order to achieve a certain level of confidence in the results.

While I definitely encourage the general idea of the authors to consider "how long measurements are long enough" and I find this to be a very relevant question, I believe they make several assumptions that may not be valid in some circumstances. This may question the robustness and generality of their methodologies, and I therefore recommend that these assumptions are discussed in much more detail and corrected wherever necessary. Below, additional explanations are given in the General Comments section, as well as some additional recommendations for further improvement.

**General comments**

1) The authors assume that the random errors are aleatoric, and completely independent. I am not certain this is the case, for example one could assume that the random error in two anemometers of the same type and on the same site may have correlation with external factors like the wind speed and turbulence. Then the random errors will be correlated to each other through the correlation to external factors. Such correlation may reduce the total variance of the difference between the errors. This issue can however be partially taken into account if the uncertainty is evaluated bin-wise as the authors are doing.

2) It is not a given that a numerical model will be able to simulate all the random variability present in real experiments. As the actual wind field in an experiment is never fully observable, we are not necessarily able to represent the entire variability in the wind conditions in the simulations, because the simulations will only vary a limited number of parameters, and the simulated wind fields will match the target statistics of the entire wind field, while the measurements are taken at a single point or a few points only. As a result, it is very often that one-to-one numerical load simulations will have smaller variance (after subtracting the bias and conditional means) than the actual measurements.

3) Power performance and load validation are addressed in the IEC 61400-12 and -13 standards respectively. Uncertainty is discussed in both standards. Any real load or power performance validation campaign would be expected to comply with the provisions of these standards. It is therefore important to align the discussions of this paper with the standards and outline any differences.

**Specific comments**

4) Page 2, line 26: I am not certain about the strict classification of bias as epistemic and the random errors as aleatoric. There may be epistemic errors that have non-zero variance (aliasing and other numerical artefacts being an example), while aleatory effects may also be autocorrelated or related to an external factor which will introduce bias for a finite experiment.

5) Equation (2): I believe this equation is true for the case when delta_P1 and delta_P2 are completely uncorrelated. This however may not be the case, for example if the error is dependent on an external parameter (say, wind speed), then delta_P1 and delta_P2 will be positively correlated and the total variance will be less than the root mean square sum that the authors suggest.

6) Page 10: the authors say "day-to-day variations should be much smaller than hour-to-hour variations". It depends – the weather systems passing over a location typically have duration in the range of 1-3 days, so there could be significant day-to-day variations. On the other hand, this may not be very important as we should not expect that "calendar days" should have influence on the external conditions, it will more likely be dependent on season and time of day.

7) As mentioned earlier, I expect that the variability produced in simulations may be smaller than what we see in real measurements, mainly because of the incomplete characterization of the true wind field passing through the rotor. I think this expectation is confirmed by the results shown in Figure 5, where there is only limited dispersion in the data.

---

## Referee Comment (RC2)

Review of "Predicting and reducing wind energy field experiment uncertainties with low-fidelity simulations" by Houck et al.

**Overall assessment**

This is a useful contribution to the often-overlooked discipline of designing experiments, here with particular relation to field testing in wind energy. The paper outlines a methodology for determining the testing duration necessary in order to obtain significant and converged results in a testing situation where a simultaneous control can be performed. This control is often possible but maybe just as often, impossible. My initial reaction was that the authors had rather neglected this latter case, giving the impression that bias uncertainties can easily be disregarded in most wind energy field experiments.

This is of course not the case. A large field experiment to examine and quantify wind farm global blockage has just been completed and here both the random and bias uncertainties are significant and both play a crucial role in determining whether a meaningful outcome of the experiment is possible.

On re-reading, maybe my initial reaction was rather harsh. In any case, I have several suggestions in the detailed comments below, to create in my view a more balanced view regarding random and bias uncertainties.

My other main comment is that the case study section is especially lengthy with a lot of detail. I am not convinced that all the detail is justified and would ask the authors to reflect on whether some simplification and shortening could be possible, for example, reducing the number of rotor size cases and possibly taking fewer Qols. This would maybe help to highlight the several important points that the case study illuminates.

**Detailed comments**

**Title**

I don't think the title quite matches the paper core topic – determining the testing duration required when random uncertainties dominate. Maybe reconsider this?

**Abstract**

L3 replace "commonly" by "sometimes" or re-word to something like "some field experiments can be conducted with a control and treatment"

**Introduction**

L24 drop the "are still of great value"?

L33 maybe add "especially where control and treatment are carried out essentially simultaneously" before "it is often safe" (or something like this...). My concern here is that if bias

errors depend critically on the conditions, in a non-simultaneous control and treatment, the bias uncertainty may still be very relevant.

L45 Probably just me but the sentence "Represented as error bars,... within the uncertainty interval." took me a long time to understand. Maybe re-word to something like "To demonstrate a significant difference, the error bars derived from the best estimate and the uncertainty at the selected confidence level, should not contain zero."

L60 "Like significance, convergence is also ensured by increasing the number of samples.." – Is significance always ensured – what if there really is no difference?

**2.1 Simulation Method**

L96 The simulation must have "acceptable accuracy" – what is this accuracy? How does this relate to the "low-fidelity" of the paper title? I think this requires a little more attention.

**2.3 Analysis and uncertainty quantification**

L166 "any bias errors should be calculated for each QoI." – I understand this as bias uncertainties (if you know the errors you can just correct for them). Please be careful in your use of "error" and "uncertainty". Could your methodology be extended to include this step – i.e. calculating an estimate of bias uncertainties based on the input (and output?) of the simulations?

**3.1 Tip Extensions**

L210 "region 2" – people not familiar with wt control may not know what this means.

**3.2 Inflow creation for the base study**

L251 "The shear exponent was then averaged for each 10-minute bin." – why not just use the 10 minute means to calculate one alpha?

L252 "It is not necessary to apply quality control to the time series" – Don't understand this. Do you mean that the necessary QC can be performed using the 10 minute statistics?

**3.5 Discussion of the Case Study**

L453 sentence ending ", perhaps the addition of a few additional samples." seem incomplete.

L466 "suss out" is rather colloquial! Maybe use boring old "determine" instead 😉.

---

## Author Comment (AC1)

Reviewers,

Thank you for your thorough and thoughtful reviews and excellent suggestions. Below you will find point by point responses to each of your comments including what or where changes to the manuscript were made in response. Your suggestions have significantly improved the manuscript specifically by making the assumptions, strengths, and limitations much clearer. We believe we have addressed all of your comments (though we recognize there will always be points within uncertainty warranting further research

*Discussion of:*

**Predicting and reducing wind energy field experiment uncertainties with low-fidelity simulations**

09 December 2023

The paper presents a virtual experiment-based methodology for estimating the amount of uncertainty in wind turbine load and power performance measurements, subsequently providing possibilities to estimate how long a measurement campaign will need to be in order to achieve a certain level of confidence in the results.

While I definitely encourage the general idea of the authors to consider "how long measurements are long enough" and I find this to be a very relevant question, I believe they make several assumptions that may not be valid in some circumstances. This may question the robustness and generality of their methodologies, and I therefore recommend that these assumptions are discussed in much more detail and corrected wherever necessary. Below, additional explanations are given in the General Comments section, as well as some additional recommendations for further improvement.

We appreciate this point. The generality was (unintentionally) overstated. We have taken effort to be more precise in our language and have addressed each specific comment as described below.

**General comments**

1) The authors assume that the random errors are aleatoric, and completely independent. I am not certain this is the case, for example one could assume that the random error in two anemometers of the same type and on the same site may have correlation with external factors like the wind speed and turbulence. Then the random errors will be correlated to each other through the correlation to external factors. Such correlation may reduce the total variance of the difference between the errors. This issue can however be partially taken into account if the uncertainty is evaluated bin-wise as the authors are doing.

Thank you for this point; it is well taken. We wrote the manuscript without clearly stating our assumption that error sources are independent. We have now updated it in several places to clarify what changes would be necessary to accommodate correlated error sources and to state when we are assuming uncorrelated errors in our analyses (see for example lines 45 and 54).

2) It is not a given that a numerical model will be able to simulate all the random variability present in real experiments. As the actual wind field in an experiment is never fully observable, we are not necessarily able to represent the entire variability in the wind conditions in the simulations, because the simulations will only vary a limited number of parameters, and the simulated wind fields will match the target statistics of the entire wind field, while the measurements are taken at a single point or a few points only. As a result, it is very often that one-to-one numerical load simulations will have smaller variance (after subtracting the bias and conditional means) than the actual measurements.

This is a good point and we have added details in regard to this in section 2.2.

3) Power performance and load validation are addressed in the IEC 61400-12 and -13 standards respectively. Uncertainty is discussed in both standards. Any real load or power performance validation

campaign would be expected to comply with the provisions of these standards. It is therefore important to align the discussions of this paper with the standards and outline any differences.

We have added a comment at the end of the section 2 and the beginning of 3.2 that acknowledges that there are standards that could be relevant to wind energy experiments and that, while the method detailed in the manuscript is not laid out to be explicitly in line with them, it is entirely adaptable to comply with them.

**Specific comments**
4) Page 2, line 26: I am not certain about the strict classification of bias as epistemic and the random errors as aleatoric. There may be epistemic errors that have non-zero variance (aliasing and other numerical artefacts being an example), while aleatory effects may also be autocorrelated or related to an external factor which will introduce bias for a finite experiment.

It seems to us that the words bias, epistemic, and "Type B" as one set and random, aleatoric, and "Type A" as another set are often used interchangeably and that it is often acceptable to do so. To clarify our meaning, we have edited the text to refer only to "bias" and "random" errors such that the proceeding text is a definition of those terms as we will use them. To further address your point about how they may be combined, we have added a clarification to say that "When they can be entirely separated…".

5) Equation (2): I believe this equation is true for the case when delta_P1 and delta_P2 are completely uncorrelated. This however may not be the case, for example if the error is dependent on an external parameter (say, wind speed), then delta_P1 and delta_P2 will be positively correlated and the total variance will be less than the root mean square sum that the authors suggest.

This is a good point and we have added a clarification regarding the possible correlation of uncertainties. Any correlation would, as you say, reduce the uncertainty of the difference relative to uncorrelated individual uncertainties.

6) Page 10: the authors say "day-to-day variations should be much smaller than hour-to-hour variations". It depends – the weather systems passing over a location typically have duration in the range of 1-3 days, so there could be significant day-to-day variations. On the other hand, this may not be very important as we should not expect that "calendar days" should have influence on the external conditions, it will more likely be dependent on season and time of day.

This is a good point and we have edited the text to say, "though day-to-day variations are likely less important than seasonal changes over weeks and months."

7) As mentioned earlier, I expect that the variability produced in simulations may be smaller than what we see in real measurements, mainly because of the incomplete characterization of the true wind field passing through the rotor. I think this expectation is confirmed by the results shown in Figure 5, where there is only limited dispersion in the data.

We believe the details added in section 2.2 also address this point in such a way as to make the reader aware of the trade-offs and possible solutions.

**Review of "Predicting and reducing wind energy field experiment uncertainties with low-fidelity simulations" by Houck et al.**

**Overall assessment**

This is a useful contribution to the often-overlooked discipline of designing experiments, here with particular relation to field testing in wind energy. The paper outlines a methodology for determining the testing duration necessary in order to obtain significant and converged results in a testing situation where a simultaneous control can be performed. This control is often possible but maybe just as often, impossible. My initial reaction was that the authors had rather neglected this latter case, giving the impression that bias uncertainties can easily be disregarded in most wind energy field experiments.

This is of course not the case. A large field experiment to examine and quantify wind farm global blockage has just been completed and here both the random and bias uncertainties are significant and both play a crucial role in determining whether a meaningful outcome of the experiment is possible.

On re-reading, maybe my initial reaction was rather harsh. In any case, I have several suggestions in the detailed comments below, to create in my view a more balanced view regarding random and bias uncertainties.

My other main comment is that the case study section is especially lengthy with a lot of detail. I am not convinced that all the detail is justified and would ask the authors to reflect on whether some simplification and shortening could be possible, for example, reducing the number of rotor size cases and possibly taking fewer QoIs. This would maybe help to highlight the several important points that the case study illuminates.

We appreciate this suggestion, but also feel that the differences in results specific to each QoI help to highlight details that the reader may not otherwise consider. For example, the way in which the results for each rotor cross each other in the edge root bending moment at wind speeds just above peak loading. These details help demonstrate the utility of the method at shining a light on specific combinations of conditions and quantities that may be especially difficult to measure.

We also added in a brief justification (~line 222) for the choice of 5 modified rotor sizes. This was to understand if would it be tremendously easier (require much less duration) to measure a 3% power gain than, say, a 1.5% power gain, for example. In the future experiment that motivated our idea for this method (the AMSIT experiment discussed in section 3.5), we need to balance an increase in power with increases in the blade loads. We wondered, if we can't run the experiment long enough to measure a difference in power, is there another QoI that would be much faster to converge. In evaluating these QoIs, we found that there are subtleties to measuring each of them, so we have shown them all for completeness. We expect that many researchers find themselves in similar dilemmas of optimizing their experiments and hope that this better justifies the length of the case study and increases its relevance.

**Detailed comments**
**Title**
I don't think the title quite matches the paper core topic – determining the testing duration required when random uncertainties dominate. Maybe reconsider this?

We agree. We have retitled the paper to "Method to predict the minimum measurement and experiment durations needed to achieve converged and significant results in a wind energy field experiment".

**Abstract**
L3 replace "commonly" by "sometimes" or re-word to something like "some field experiments can be conducted with a control and treatment"

Agreed and done.

**Introduction**
L24 drop the "are still of great value" ?

Agreed and done.

L33 maybe add "especially where control and treatment are carried out essentially simultaneously" before "it is often safe" (or something like this..). My concern here is that if bias errors depend critically on the conditions, in a non-simultaneous control and treatment, the bias uncertainty may still be very relevant.

We have included your suggestion and also clarified that binning of data by atmospheric conditions should help for non-simultaneous measurements. We have also made similar edits in other places in the manuscript where this comes up.

L45 Probably just me but the sentence "Represented as error bars,... within the uncertainty interval." took me a long time to understand. Maybe re-word to something like "To demonstrate a significant difference, the error bars derived from the best estimate and the uncertainty at the selected confidence level, should not contain zero."

We have reworded this for clarity.

L60 "Like significance, convergence is also ensured by increasing the number of samples.." – Is significance always ensured – what if there really is no difference?

We see how this is confusing. If there is no difference, zero will always fall within the interval represented by the errorbars. As more samples are added, the interval will shrink, but there will continue to be no difference and this wouldn't be referred to as a significant non-difference. We have amened this to say, "Convergence is ensured by increasing the number of samples..."

**2.1 Simulation Method**

L96 The simulation must have "acceptable accuracy" – what is this accuracy? How does this relate to the "low-fidelity" of the paper title? I think this requires a little more attention.

We have added this statement: "This requires expert judgement to ensure the model fidelity does not neglect effects critical to the measurement of interest. For example, if the three-dimensional flow around the blades is considered important to the QoI, then a Blade Element Momentum approach may not suffice."

**2.3 Analysis and uncertainty quantification**

L166 "any bias errors should be calculated for each QoI.." – I understand this as bias *uncertainties* (if you know the errors you can just correct for them). Please be careful in your use of "error" and "uncertainty". Could your methodology be extended to include this step – i.e. calculating an estimate of bias uncertainties based on the input (and output ?) of the simulations?

You are right! We were being too loose in our use of "error" and "uncertainty" and have revised this throughout the manuscript. Regarding L166, calculation of uncertainties due to bias errors should be straightforward if they are known, which is one reason we chose not to explicitly demonstrate how it would be done. To be most accurate, however, it may have to include biases introduced in modeling, which would be uncovered during a thorough validation of the turbine model, and such data are rarely available. We have added some text to better explain this subtlety at the end of section 2.3.

**3.1 Tip Extensions**

L210 "region 2" – people not familiar with wt control may not know what this means.

This has been edited to say (line 232), "...to ensure that it is operated optimally by finding the combination of blade pitch and tip speed ratio that maximizes $C_P$ in region 2 of the power curve. Region 2 is defined as the range of wind speeds where the turbine produces power but wind speeds are too low for it to reach rated power."

**3.2 Inflow creation for the base study**

L251 "The shear exponent was then averaged for each 10-minute bin." – why not just use the 10 minute means to calculate one alpha?

Given the temporal variability in the flow, I suspect that you get a "truer" average of the shear by calculating it for all profiles in time and then averaging. That said, you might get a better fit by using the average profile, as you suggest, because it ensures that any extreme or more difficult to fit profiles in time do not throw off the average. We have added text (~ line 275) to this point as it may be more appropriate for some data sets and clarifies that alternative choices can be made at various steps while still using the general method to predict the necessary durations.

L252 "It is not necessary to apply quality control to the time series" – Don't understand this. Do you mean that the necessary QC can be performed using the 10 minute statistics?

We have clarified this (now at ~ line 277).

**3.5 Discussion of the Case Study**

L453 sentence ending ", perhaps the addition of a few additional samples." seem incomplete.

This sentence has been fixed.

L466 "suss out" is rather colloquial! Maybe use boring old "determine" instead .

Agreed and changed.

---

## Referee Report (RR1)

*Discussion of:*

**Method to predict the minimum measurement and experiment durations needed to achieve converged and significant results in a wind energy field experiment – Revision 1**

11 March 2024

This is a revised version of a paper that presents a virtual experiment-based methodology for estimating the amount of uncertainty in wind turbine load and power performance measurements, subsequently providing possibilities to estimate how long a measurement campaign will need to be in order to achieve a certain level of confidence in the results.

With their revision, the authors have attempted to address all reviewer comments and have satisfactorily addressed most of them. There are however a couple of items where I believe a bit more efforts are necessary. These are listed below:

1) It is great that the authors did add a mention of the IEC standards, saying that their approach is fully adaptable to the standards. I think however that they need to go a bit further and directly discuss how their approach should be adapted to the IEC. The vast majority of such test campaigns are for certifying a certain product or functionality, and then compliance with the standard is a requirement which will take precedence over many other considerations.

2) Section 2.2: as recommended, the authors have included statements regarding how simulations do not represent the full variability of the inflow. While this is good to start with, I do not fully agree with the example given by the authors. While TurbSim indeed will drive the distribution of wind towards Gaussian, I believe increasing the simulation period above 10 minutes will lead to relatively small changes in the variability of the flow, because the Veers turbulence model (and other similar models like the Mann model) do not have physical mechanisms that produce turbulence energy with low frequencies. If the tails are longer this is mainly due to more data, the distribution still being Gaussian. This is contrary to physical measurements and more advanced simulation frameworks such as LES. Therefore, more simulations per bin or longer simulations will generate some variability, but not all that is present in the measurements. Another major variability we see in measurements is due to the uncertainty in the inflow characterization, because the wind field is not fully observable (the cup anemometer may have measured 9.8m/s, but the mean speed over the entire rotor may have been 10.5m/s for example). Please discuss this rather than the Gaussianity of the flow which I think has little effect here. How can the lack of wind measurement uncertainty affect the validity and the usability of the outcomes of a virtual experiment? This is the important question that should be addressed.

---

## Author Response (AR2)

Thank you for the additional feedback. It was very well received and we agree that additional details were needed in those two section. Below, please find the edits that we have made beginning a little before the sections mentioned for complete context.

**Reviewer comment:**
It is great that the authors did add a mention of the IEC standards, saying that their approach is fully adaptable to the standards. I think however that they need to go a bit further and directly discuss how their approach should be adapted to the IEC. The vast majority of such test campaigns are for certifying a certain product or functionality, and then compliance with the standard is a requirement which will take precedence over many other considerations.

**Response:**
The method described and demonstrated herein is highly flexible and adaptable to the particular needs of the experiment. At a very high level, it consists of performing a suite of simulations to represent a proposed experiment with a balance between computational time and fidelity. The outputs of the simulations are then used to perform a statistical analysis to quantify uncertainty and convergence to standards determined by the user and this data is finally converted into a prediction of the minimum measurement and experiment durations required to produce significant and converged results. At this level, the proposed method could be used for a variety of experiments in many fields, though the focus here is on wind energy and, in particular, field experiments as these present a particular challenge with long measurement durations required to reduce uncertainty due to random errors.

It should also be acknowledged that there are IEC standards relevant to wind energy field experiments \citep[][]{61400-12,61400-13} that researchers may choose to follow. The method laid out herein does not explicitly follow these standards, but it is entirely adaptable to comply with them. If, for example, one wished to follow IEC 64100-12-1 to create a power curve according to standards, then it would be necessary to use the method of bins for uncertainty analysis with the simulated data as detailed in Annex E of that standard. As this is a virtual experiment, however, some assumptions may need to be made regarding the many sources of uncertainty that are tracked and included by the standard but that are not explicitly represented in the virtual experiment. The Category B uncertainties in IEC 61400-12-1 could be to help define an appropriate range of simulation input parameters, for example on wind speed, shear, air density, etc. Uncertainties that cannot be included in estimating input parameters can be included in post-processing of the data. In fact, by including reasonable estimates of every source of uncertainty,, it would be possible to rank the importance of each source through an uncertainty quantification and thereby determine which may be most critical to reduce.

**Reviewer comment:**
Section 2.2: as recommended, the authors have included statements regarding how simulations do not represent the full variability of the inflow. While this is good to start with, I do not fully agree with the example given by the authors. While TurbSim indeed will drive the distribution of wind towards Gaussian, I believe increasing the simulation period above 10 minutes will lead to relatively small changes in the variability of the flow, because the Veers turbulence model (and other similar models like the Mann model) do not have physical mechanisms that produce turbulence energy with low frequencies. If the tails are longer this is mainly due to more data, the distribution still being Gaussian. This is contrary to physical measurements and more advanced simulation frameworks such as LES. Therefore, more simulations per bin or longer simulations will generate some variability, but not all that is present in the measurements. Another major variability we see in measurements is due to the uncertainty in the inflow characterization, because the wind field is not fully observable (the cup anemometer may have measured 9.8m/s, but the mean speed over the entire rotor may have been 10.5m/s for example). Please discuss this rather than the Gaussianity of the flow which I think has little effect here. How can the lack of wind measurement

uncertainty affect the validity and the usability of the outcomes of a virtual experiment? This is the important question that should be addressed.

**Response:**

After selecting the simulation method and having acquired representative inflow data, the inflow data are now processed into the format required by the simulation code. Here, the method uses 10-minute bin intervals, which is standard for wind energy field experiments, though it could be easily adapted for other needs. This accepts that the effects of phenomena happening on shorter time scales could be reduced due to long averages and phenomena happening on longer time scales may not be adequately captured, so this averaging time is an important consideration depending on the goals of the experiment. Indeed, numerical representations of inflows will almost certainly underrepresent the true variability in the inflow. TurbSim, for example, will drive the velocity distribution toward a Gaussian and longer simulation times generally create longer tails within the extremes that the model can capture, which will to a point  capture a more complete representation of the inflow. If the QoI is an extreme that the model can capture, say, a maximum load, then bins longer than 10 minutes may be necessary such that this QoI is recorded relative to the mean conditions upon binning by condition (binning by condition will be discussed below). If, however, average quantities are of interest, then more 10-minute bins will generally help make up for missing the tails of the distributions of any inflow parameters in each bin.

While more simulations per bin and/or longer simulations will help to replace some of the variability missed when comparing modeled inflows to measurements, it will not close the gap entirely. As mentioned in section \ref{SimMeth}, the proposed method will only yield meaningful results if the modeling tools can capture the QoI, which will require input from subject matter experts. If the QoI is believed to be sensitive to inflow fidelity, then comparisons could be made against higher fidelity methods, such as large-eddy simulation (LES), to verify the adequacy and/or quantify the uncertainty of the low-fidelity approach. These uncertainties can then be incorporated into the final analysis.

Some uncertainties, however, such as the difference between measurements at the met tower and conditions at the rotor are important to retain in the virtual experiment as they can help replicate the real experiment. For example, the velocity measured at the met tower may be biased from the velocity at the rotor. In the control and treatment scenario presented here, this bias is inherently subtracted out. When there is not an available control, such biases in measurements would be critical to capture in the simulations or to incorporate into the post-processing and analyses of the data. Representations of uncertainties in the inflow measurements themselves can and should be included in the uncertainty analysis of the virtual experiment.